# Prospective contributions of biomass pyrolysis to China's 2050 carbon reduction and renewable energy goals

Qing Yang [1,2,3,4,10 ✉], Hewen Zhou[1,3,10], Pietro Bartocci [5], Francesco Fantozzi [5], Ondřej Mašek[6], Foster A. Agblevor[7], Zhiyu Wei[3,4], Haiping Yang[1,3,4], Hanping Chen[1,3,4 ✉], Xi Lu [8], Guoqian Chen [9], Chuguang Zheng[1,3], Chris P. Nielsen[2] & Michael B. McElroy [2 ✉]

Recognizing that bioenergy with carbon capture and storage (BECCS) may still take years to mature, this study focuses on another photosynthesis-based, negative-carbon technology that is readier to implement in China: biomass intermediate pyrolysis poly-generation (BIPP). Here we find that a BIPP system can be profitable without subsidies, while its national deployment could contribute to a 61% reduction of carbon emissions per unit of gross domestic product in 2030 compared to 2005 and result additionally in a reduction in air pollutant emissions. With 73% of national crop residues used between 2020 and 2030, the cumulative greenhouse gas (GHG) reduction could reach up to 8620 Mt $CO_2$-eq by 2050, contributing 13–31% of the global GHG emission reduction goal for BECCS, and nearly 4555 Mt more than that projected for BECCS alone in China. Thus, China's BIPP deployment could have an important influence on achieving both national and global GHG emissions reduction targets.

[1] State Key Laboratory of Coal Combustion, Huazhong University of Science and Technology, Wuhan, PR China. [2] John A. Paulson School of Engineering and Applied Sciences, Harvard University, Cambridge, MA, USA. [3] Department of New Energy Science and Engineering, School of Energy and Power Engineering, Huazhong University of Science and Technology, Wuhan, PR China. [4] China-EU Institute for Clean and Renewable Energy, Huazhong University of Science and Technology, Wuhan, PR China. [5] Department of Engineering, University of Perugia, Perugia, Italy. [6] UK Biochar Research Centre, School of GeoSciences, University of Edinburgh, Edinburgh, UK. [7] USTAR Bioenergy Center, Department of Biological Engineering, Utah State University, Logan, UT, USA. [8] School of Environment and State Key Joint Laboratory of Environment Simulation and Pollution Control, Tsinghua University, Beijing, PR China. [9] College of Engineering, Peking University, Beijing, PR China. [10] These authors contributed equally: Qing Yang, Hewen Zhou. ✉email: qyang@hust.edu.cn; hp.chen@163.com; mbm@seas.harvard.edu

To meet the targets agreed in the Paris Agreement to limit global temperature increases to 2 °C or possibly 1.5 °C[1–3] cost-effectively, widespread applications of negative emission technologies (NETs) are considered essential. Options include two main NETs relying on photosynthesis: (1) bioenergy with carbon capture and storage (BECCS) and (2) production and sequestration of biochar[4–6]. For the long-term global temperature target, BECCS was the only significant NET considered in the scenarios developed in the IPCC Fifth Assessment Report[7]. However, this option is developing slowly and needs years to mature. Time is running out as deployment of NETs needs to start essentially immediately if we are to reduce the risk of overshooting international carbon goals, avoid excessive costs of mitigation, and minimize negative climate-related effects[8–10]. This study presents a ready-to-implement biochar technology based on pyrolysis of biomass, followed by sequestration of biochar in soil and use of associated biofuels as an energy source and chemical feedstock, offering a near-term technology option that can be applied until widespread BECCS deployment becomes more feasible[11–13]. It features high GHG reduction intensities (see Table 1), economic benefits (including avoidance of costly CCS facilities), environmental benefits (to air quality and soil quality), and a wide range of available operational scales[14–16].

Biomass pyrolysis is an established thermochemical technology converting biomass into three main product streams: biochar, pyrolysis gas, and bio-oil[17]. Biochar is usually mixed with fertilizer and then returned to soil, serving as both a medium for carbon storage and as a beneficial amendment to soils (explained in Supplementary Note 12–13 and Supplementary Fig. 5). Pyrolysis gas and bio-oil are fuels: pyrolysis gas can be used to generate electric power and as an energy source for household heating and cooking and other applications, while bio-oil can also be used as a fuel but is most valuable as an alternative to coal tar as a feedstock in the chemical industry. (Note that biomass

pyrolysis also produces commercial bio-acid, but in quantities too minor for strong consideration in this paper.)

There are three main types of biomass pyrolysis: fast, intermediate, and slow[18]. This paper focuses on China and a state-of-the-art biomass intermediate pyrolysis poly-generation (BIPP) system as a near-term alternative to BECCS until the latter becomes more feasible. The term "poly-generation" refers to multiple outputs of an integrated technology system, which in the current case will refer to a pyrolysis reactor coupled with a heat recovery system, a microgenerator to burn 80% of the biofuels to generate electric power, as well as a storage and distribution system to supply the remaining 20% of biofuels for household use (see in Supplementary Note 2). This BIPP system has a number of advantages in terms of technical[19], economic[20], and potential environmental performance[21]. Technically, compared to fast pyrolysis, for which the dominant product is bio-oil, intermediate pyrolysis operates with a residence time of 30 min at a 600 °C optimal working temperature, which enables a full secondary reaction[20–22] and is capable thus of yielding relatively more biochar (33–37% increased yield). Compared to slow pyrolysis, for which the primary product is biochar, intermediate pyrolysis poly-generation provides considerable heat and opportunities for generation of electricity through production of pyrolysis gas and bio-oil; it also produces biochar with comparatively high stability (stable carbon percentages in soil by weight of 60–80% over a 100-year period)[23–25] in a continuous production system. Furthermore, the system has good feedstock flexibility, which means it is well suited for the diverse range of biomass sources available in China, offering the potential to deploy a variety of pyrolysis products[19]. Indeed, the first successful deployments of this technology have already occurred in China[26] (detailed descriptions in Supplementary Note 7). Moreover, the applications of BIPP systems have potentially positive environmental effects especially on the reduction of carbon levels in the atmosphere and on soil management, the latter achieved by using the biochar as a

**Table 1 Comparison of GHG emissions reduction (including fossil fuel offset) for pyrolysis systems with sequestration or use of biochar as fuel and use of other biofuels and products.**

| Study | Process | Product applications | GHG emissions reduction intensity (g $CO_2$-eq $MJ^{-1}$) |
|---|---|---|---|
| This study | Biomass pyrolysis poly-generation system Temperature: 600 °C Residence time: >1800 s | Biochar: soil application Pyrolysis gas: substitution of coke oven gas and electricity production Bio-oil: substitution of coal tar in chemical raw materials | 136.45 |
| | | Biochar: charcoal substitution in industries Pyrolysis gas: substitution of coke oven gas and electricity production Bio-oil: substitution of coal tar in chemical raw materials | 46.80 |
| Peters et al.[28] (Spain) | Biomass slow pyrolysis system Temperature: 450 °C Residence time: ~2500 s | Biochar: soil application Pyrolysis gas: heat production for pyrolysis system and substitution of natural gas Bio-oil: heat production for pyrolysis system | 122.18 |
| | | Biochar: charcoal substitution in coal power plant Pyrolysis gas: heat production for pyrolysis system and substitution of natural gas Bio-oil: heat production for pyrolysis system | 63.22[a] |
| Roberts et al[14]. (the United States) | Biomass slow pyrolysis system Temperature: 450 °C Residence time: long enough | Biochar: soil application Pyrolysis gas: substitution of natural gas for heat product | 108.57 |
| | | Biochar: charcoal substitution in IGCC plant Pyrolysis gas: substitution of natural gas for heat product | 36.64 |

[a]The reference does not consider the GHG emissions derived from construction process (e.g., equipment and installation) in life-cycle assessment.

soil amendment. To sum up, BIPP systems can provide a triple benefit in the near term not only in addressing climate change but as an environmentally advantageous source of energy and as a source for improvement of the quality of soils. Despite a great deal of experimental research and early commercial applications in China, a comprehensive assessment of the potential for BIPP systems in the context of GHG emissions mitigation is not as yet available. There is also no specific policy to develop and deploy BIPP technologies in China. This information is critical for a more complete debate on climate change mitigation options, as well as for planning future energy and climate policies.

To fill gaps in current knowledge of BIPP systems and their role in climate change mitigation[27–29], a dynamic hybrid assessment model (Aspen Plus© combined with hybrid life-cycle assessment) has been developed incorporating a wealth of experimental data (e.g., the O:C ratio for biochar produced from different crop residues) and operational information derived from demonstration pyrolysis plants. The trade-offs among technical, economic, and environmental effects of BIPP systems are explored. Detailed analysis of carbon mitigation potentials and air quality benefits at the provincial level are provided, taking account of crop types, selling prices and yields, as well as soil characteristics in different provinces. Moreover, different scenarios of deployment of bioenergy NETs (BIPP and/or BECCS) are explored to identify the best route to accelerate near-term GHG emission reductions and meet mid-century emission targets. In this context, the results of this work should fill a gap on a topic that is urgent but has been insufficiently examined to date[30].

The analysis shows that by controlling variables for BIPP systems, notably the temperature of pyrolysis and the production capacity, positive financial returns can be achieved from production of biofuels along with significant climate mitigation benefits as the carbon-rich biochar is applied to soil. A sustained national investment in biochar production to exploit 33% of sustainable available crop residues in China could potentially reduce China's GHG emissions by as much as 54.27 Mt $CO_2$-eq per year. Meanwhile political support for subsidies would be needed to develop BIPP in provinces with relatively poor economic performance, such as Henan and Shanxi. In addition, the results suggest that significant benefits for air quality could be achieved by deploying BIPP systems on a national scale, especially in developed eastern coastal areas burdened with high levels of air pollution. Evaluating the effects on future GHG emissions, under a "moderate" scenario that assumes processing 73% of crop residues into biochar and biofuels using BIPP in the near term (2020–2030) and then coordinated deployment with BECCS after 2030, it is concluded that the cumulative GHG emissions reduction could reach as high as 8620 Mt $CO_2$-eq by 2050. A detailed, conservative analysis suggests that this reduction would account for 13–31% of the global GHG emission reduction goal for BECCS (28–65 Gt $CO_2$-eq, in the 1.5 °C pathway in IPCC's special report[31]) by 2050. On the other hand, relying exclusively on BECCS technology that might become feasible after 2030 (as discussed in the scenario analysis below) would contribute to a reduction of no more than about 4066 Mt $CO_2$-eq by 2050. In a set of "maximum" scenarios, which assume that all conceivably available biomass (not only crop residues but also energy crops and forest residues) is used to produce biochar and biofuels, the GHG reduction could reach as much as 3156 Mt $CO_2$-eq and 21,803 Mt $CO_2$-eq by 2030 and 2050, respectively. Its national deployment could contribute to a reduction of as much as 61% of carbon emissions per unit of GDP in 2030 compared to 2005[32]. Indeed, this reduction by itself could have a significant impact on achievement of the goals (i.e., 60–65% $CO_2$ intensity reduction) included in China's Nationally Determined Contribution (NDC) to the Paris Agreement[33]. Thus, China's BIPP deployment could

have an important influence on achieving the national near-term (2030) and global mid-century (2050) GHG emissions reduction targets.

## Results

**Dynamic simulation of a BIPP system.** The energy and mass balance of a BIPP system depends on a number of factors including the biomass feedstock type, heating rate, and reaction time, in addition to the peak pyrolysis temperature (the key parameter considered in this study, as explained in Supplementary Note 3). Based on the composition parameters of eight major crop residue types (accounting for more than 90% of crop residues) in China, a model was built and used for calculations with the Aspen Plus simulator. The material flows explored in this analysis include the cooling water and the biomass fuel used for heating the pyrolysis reactor. The heat required by the biomass pyrolysis reactor, provided by high-temperature flue gas generated by combustion of part of the biomass feedstock, increases with increasing pyrolysis temperature, as shown in Fig. 1a. The figure indicates that the fuel demand rises at a faster rate in the pyrolysis temperature range 250–450 °C as a result of the decomposition of cellulose and hemicellulose. Above 450 °C, the slow rate of increase in fuel consumption is related mainly to the increase in temperature and the slow decomposition of lignin.

The demand for cooling water is associated mainly with the yield of biochar (see Fig. 1a) and the difference between the cooling (150 °C) and pyrolysis temperatures. Thus, the trend in cooling water demand is not monotonic with the increase in pyrolysis temperature as explained in Supplementary Note 5.

The detailed energy flow analysis is outlined in Supplementary Note 6, including energy performance and the energy flow diagram. The results show that the energy efficiency declines gradually with the increase in pyrolysis temperature. It is noted that the consumption of fuel causes a large dissipation of heat in

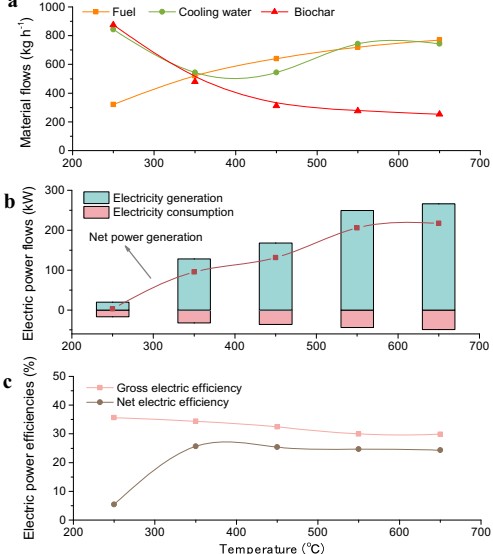

**Fig. 1 Performance of a BIPP system with temperature varying from 250 to 650 °C for 1 t biomass input per hour. a** Material flows: fuels input for high-temperature flue gas generation, cooling water inputs used to cool biochar, and the biochar output; **b** electric power flows: electricity generation by the pyrolysis gas combustion power system, electricity consumption of the whole system, and net power generation; **c** electric power efficiencies: gross/net efficiencies of electricity generation in the BIPP system, which are defined as the gross/net power generation divided by the energy input from pyrolysis gas.

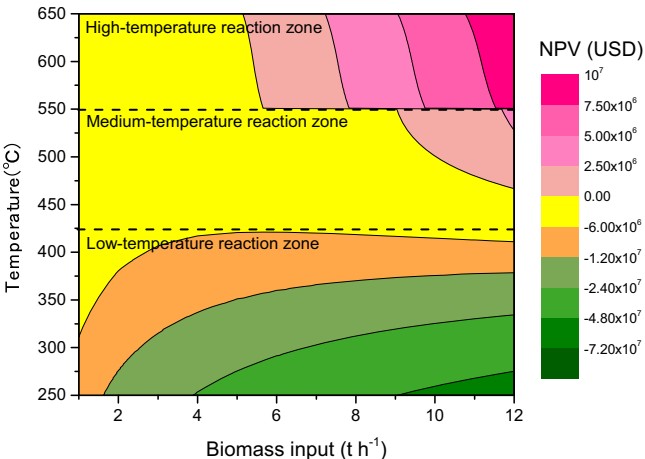

**Fig. 2 The two-factor sensitivity analysis for net present value.** Note that there is a step change at 550 °C caused by the pyrolysis gas price change in different regimes of heating value (shown in Supplementary Note 7).

the whole system (Supplementary Fig. 4). The results in Fig. 1b indicate that power consumption has an upward trend with increase in pyrolysis temperature in accord with the trend of fuel consumption in the combustion unit. However, when the pyrolysis temperature is above 350 °C, the increase seems to be less linear over the entire range. On the other hand, the potential power generation by the system increases with pyrolysis temperature, following the yield of pyrolysis gas (see Supplementary Fig. 3). Overall, the net power output of BIPP increases with rising temperature. The gross and net electric efficiencies are stable between 350 and 650 °C (see Fig. 1c), but the net electric efficiency shows a large increase between 250 and 350 °C that is mainly related to changes of auxiliary power required by the integrated BIPP systems.

**Economic feasibility for BIPP with biochar incorporated in soil and biofuels used for energy and as a chemical feedstock.** The scale (biomass feedstock capacity) and operating temperature of the poly-generation system represent the two most important factors affecting the project's revenue, which is derived chiefly from the co-production with biochar of commercially valuable pyrolysis gas as well as bio-oil[34]. Figure 2 illustrates the influence of the key factors for the net present value (NPV) of BIPP. The result indicates that the reaction temperature and biomass feedstock capacity work significantly together to determine the system's revenue. For the purpose of this analysis the reaction temperature range was divided into three zones according to the trend of economic benefit: low (250–420 °C), medium (420–550 °C), and high (550–650 °C). For the low- and medium-temperature zones, the NPV of the BIPP system is always negative, but the low-temperature zone is the most sensitive to changes in biomass input. For both the low- and medium-temperature zones, the NPV decreases with the increase in biomass feedstock input. On the other hand, in the high-temperature zone the NPV increases, achieving consistently positive NPV for biomass throughputs over 6.0 t/h.

Overall, from a techno-economic perspective, it is suggested that this kind of BIPP system with biochar returned to soil and biofuels used for direct energy use, power generation, and industrial use should be developed on a relatively large scale with higher pyrolysis temperatures between 550 and 650 °C. There also are studies showing that the variation of pyrolysis parameters could influence the profitability of the process technology[35,36]. To date, there has been a large number of published studies

discussing properties of biochar and biofuels produced at pyrolysis temperatures of 300–700 °C[37,38]. These studies have shown that, from an economic perspective, BIPP systems operating at higher temperatures (above 550 °C) perform better. Thus, the results of the economic analysis indicate that the proposed novel BIPP system with biochar sequestration can be profitable in a mature carbon market without subsidies.

**Life-cycle GHG emissions for a demonstration BIPP system.** Based on the techno-economic analysis results above, a demonstration BIPP system operating at pyrolysis temperatures between 525 and 650 °C was selected and employed to evaluate associated net life-cycle GHG emissions. The net life-cycle GHG emissions define the life-cycle GHG emissions associated with crop cultivation (e.g., soil carbon loss, fertilizer utilization), transportation and the production processes of material inputs, subtracting the sum of carbon fixation from carbon sequestered in the form of biochar in soil and reduced GHG emissions from the biochar soil effect (explained in "Methods"). $CO_2$ emissions from the combustion of biomass feedstock, bio-oil, and pyrolysis gas are treated as net zero, assuming that the $CO_2$ released is captured during the biomass growth cycle. The life-cycle GHG emissions have been calculated for each subsystem, including contributions from agricultural production, transportation (for biomass feedstock and biochar application, see Supplementary Note 11), plant construction, and operation and maintenance (O&M, for the pyrolysis system and for the pyrolysis gas combustion power generation system), as well as water treatment that has been mostly overlooked in previous studies (Supplementary Note 11). Results (Fig. 3b) show that on average the largest contribution to the system GHG emissions is related to plant construction (42%), followed by O&M (25%), water treatment (13%, relating mainly to production of $CH_4$ from wastewater), agricultural processes (14%), and transportation (6%). A detailed breakdown for agricultural processes is shown in Fig. 3c. The figure shows that indirect emissions from fertilizers (especially N- and K-based ones), and direct GHG emissions (t $CO_2$-eq) from the change in soil organic carbon (SOC) and fertilizer use are the most important factors, contributing to release of 649 t $CO_2$-eq and 457 t $CO_2$-eq per year, respectively.

Furthermore, biochar stability can influence its carbon sequestration potential[39–41]. It is estimated that the stability of biochar[14,27] (as shown in Supplementary Table 16) ranges from 60 to 80% (73% for the demonstration BIPP system) over a time span of 100 years, based on experimental studies exploring the sensitivity of results to the O:C ratio of the biochar, which is determined largely by the types of biomass feedstock, processing, and environmental conditions (as reported in the Supplementary Note 12). The net life-cycle GHG emissions per unit of energy output (including electricity, pyrolysis gas and bio-oil, units for which are converted into MJ) ranged from −60.58 to −12.49 g $CO_2$-eq MJ$^{-1}$ (see Fig. 3a), depending on the temperature and stability of biochar in the poly-generation plant. The lowest net life-cycle GHG emissions are realized at a temperature of around 600 °C. At the same time, the stability of biochar equal to 73% in the demonstration BIPP system can reach a peak of about −51.38 g $CO_2$-eq MJ$^{-1}$. These results confirm that with a temperature range between 550 and 650 °C, which has been shown to be economically favorable, the poly-generation system can achieve negative GHG emissions with a 100-year carbon stability of between 60 and 80%.

The total GHG emissions reduction is the value of the GHG emissions offset through substitution of fossil fuels with pyrolysis products (see Table 1), subtracting the net life-cycle GHG emission. Finally, BIPP plants with biochar sequestration can thus

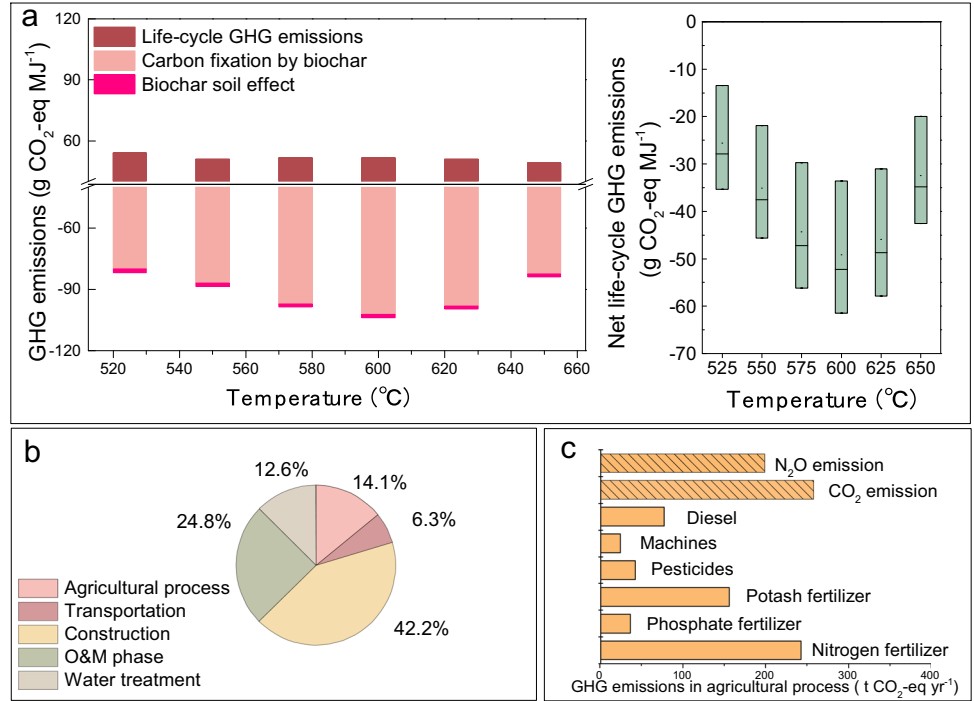

**Fig. 3 The GHG emissions for a demonstration BIPP system with biochar sequestration (excluding fossil fuel offset). a** Left figure presents life-cycle GHG emissions, carbon fixation by sequestering biochar into soil (influenced by the stability of biochar), and reduced emissions by biochar soil effect (explained in "Methods"), while the right figure represents the net life-cycle GHG emissions for the whole system; **b** the GHG emission shares of five subsystems at 600 °C with the lowest net life-cycle GHG emissions; **c** the GHG emissions for different components of agricultural production (with $N_2O$ and $CO_2$ emissions coming from soil as explained in Supplementary Note 13).

result in GHG emission reductions of 136.45 g $CO_2$-eq $MJ^{-1}$. This result represents an increase of 191.56% in the reduction of GHG emissions compared to an alternative scenario in which biochar is instead used as a substitute for charcoal as solid fuel in industry market (yielding a GHG emission reduction of 46.80 g $CO_2$-eq $MJ^{-1}$, calculated according to emission factors in Supplementary Table 21, see Table 1; this alternative use of biochar is thus not considered further in this paper). In addition, comparison of the present results with other studies shows that slow pyrolysis systems (with an almost pure biochar product) are capable of achieving GHG emissions reductions ranging from 108.57 to 122.18 g $CO_2$-eq $MJ^{-1}$ through biochar sequestration (larger than biochar product substitution effects). This indicates that biochar sequestration coordinated with substitution of by-products (pyrolysis gas and bio-oil) in BIPP in China could bring about greater reductions in potential GHG emissions. Thus, compared with conventional biomass technologies, BIPP systems with biochar sequestration could result in large GHG emissions reduction and even negative GHG emissions in China.

**Spatial distribution of reduction of GHG emissions and economic feasibility of to the deployment of BIPP on a national scale.** To ensure that this study can estimate comprehensively the reduction in national GHG emissions, the calculations for the spatial distributions of biomass and soil types combined with experimental results have been taken into account. Eight typical agricultural residues in China have been considered with various selling prices in 31 provinces in China. Eight kinds of biochar are thus assigned with different carbon stabilities. In Fig. 4, the provincial GHG emissions reductions per year are evaluated assuming deployment of BIPP units (based on a sustainable 33% of all available biomass, as explained in Supplementary Note 14). BIPP systems and applications of biochar to soil have been shown

to have a number of positive effects, including reduction in agricultural $N_2O/CH_4$ emissions (as a result of biochar serving as a soil amendment), substitution of biofuels as alternatives for fossil fuels and potentially most importantly, an increase in soil carbon storage. The results clearly indicate that provinces rich in biomass resources could have a significant potential for GHG emission reductions, since the distribution of the biomass resource mainly affects the number of installed pyrolysis plants, and further promotes GHG emission reductions from fossil fuel substitution in addition to allowing for the fixation of carbon as biochar into soils (see Supplementary Fig. 7). However, the results may also vary in response to changes in crop types, an important factor in life-cycle GHG emissions for biomass utilization technology[39]. In the agricultural process, various factors (e.g., biomass availability and types, soil types, and average temperature distributions) were considered in the detailed calculations (see in Supplementary Note 11–13 and Supplementary Figs. 6 and 7). As illustrated in Supplementary Fig. 7, generally high GHG emissions are attributable to SOC loss, for example in northeast China and Inner Mongolia (with the most fertile soils in China). In addition, crop types such as rice, which is a considerable source of $CH_4$ emissions, drive GHG emissions to some extent in southern provinces such as Anhui and Jiangxi. In the soil sequestration process, the factors including crop types and pyrolysis conditions for carbon storage of biochar were considered based on experimental results (shown in Supplementary Table 16 and Fig. 4c). As a result, the reduction in national GHG emissions could reach as high as 54.27 Mt $year^{-1}$. BIPP deployment can have a particularly important influence in central eastern China, which is responsible for notably high GHG emissions (Fig. 4b).

In addition, the economic indicator of NPV for BIPP systems has been studied as a function of different provincial biomass conditions (explained in Supplementary Note 9). It is found that the selling price of crop residues plays a significant role in

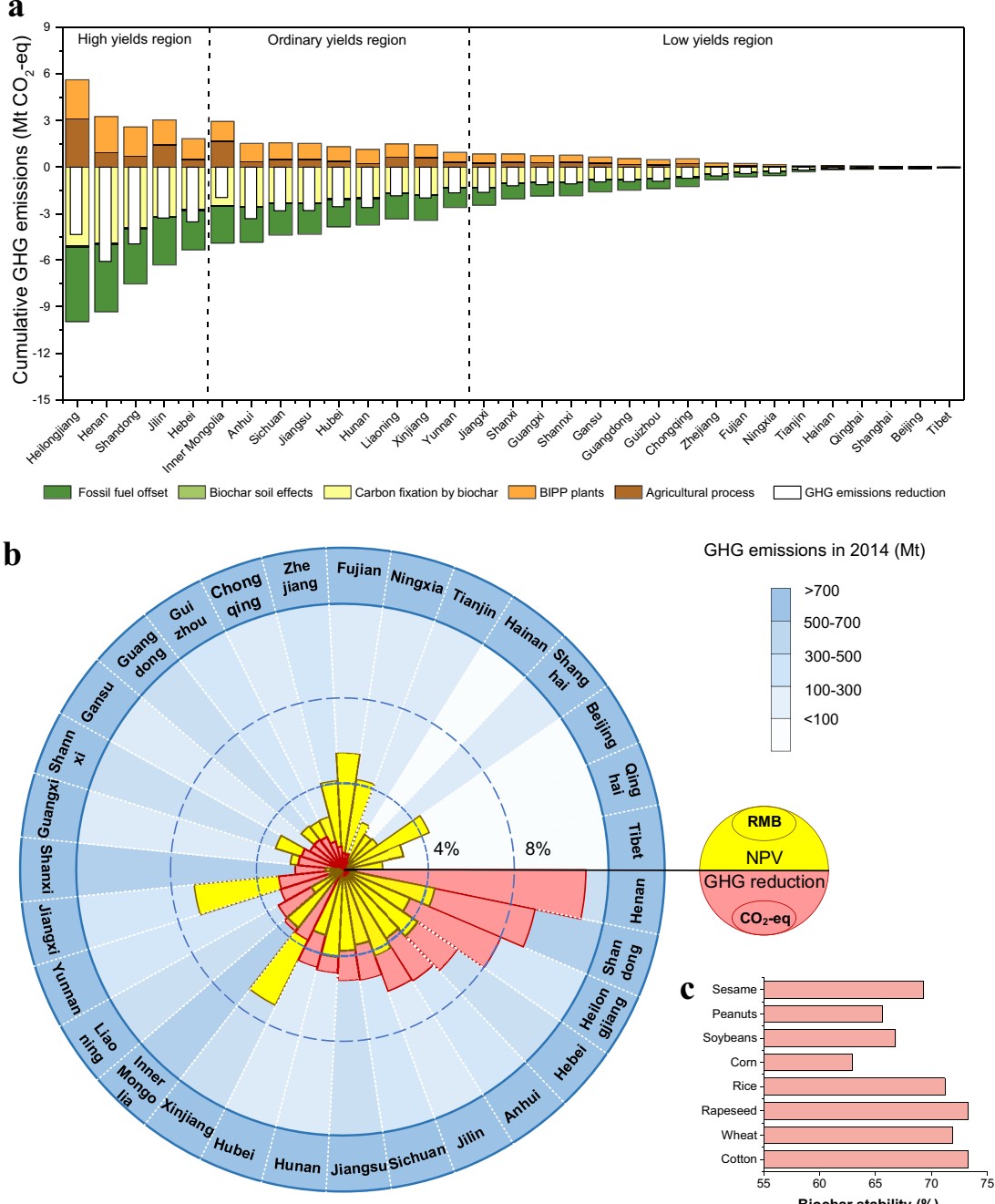

**Fig. 4 Spatial distribution of GHG emissions reduction and economic feasibility of national deployment of BIPP systems. a** Comparison of GHG emissions reductions across Chinese provinces with deployment of BIPP systems (note that provinces are ordered by the amount of available crop residues); **b** overlapping graph of shares (%) of national GHG emissions reduction per year (red sectors) that could be achieved in Chinese provinces using BIPP systems with biochar sequestration (omitting Taiwan, Macao, and Hong Kong), and shares (%) of national NPV economic performance (yellow sectors); the blue background shades indicate total GHG emissions levels in 2014[32], as defined at right; **c** the stability of biochar produced from various crop residues of a BIPP system at 600 °C.

economic performance. For example, a BIPP system in Henan Province has a poor NPV because of the relatively high price of wheat and rice crops although there is abundant biomass. As a result, provinces such as Shandong, Hebei, and Jilin could develop BIPP technology with strong economic feasibility (Fig. 4b).

**Air quality benefits due to nationwide deployment of BIPP with biochar sequestration.** Besides the CO₂ mitigation benefits

offered by deployment of BIPP systems, there is also considerable potential for reduction of air pollution, contributing to China's near-term air quality goals. In this study, three potential pathways are considered for air pollution reduction as a result of the assumed deployment of BIPP systems, including: substitution effects due to the use of pyrolysis products displacing traditional emission sources such as thermal power generation and coke oven gas and coal tar use (less new air pollution emissions from the combusted biomass stream used to generate heat for pyrolysis

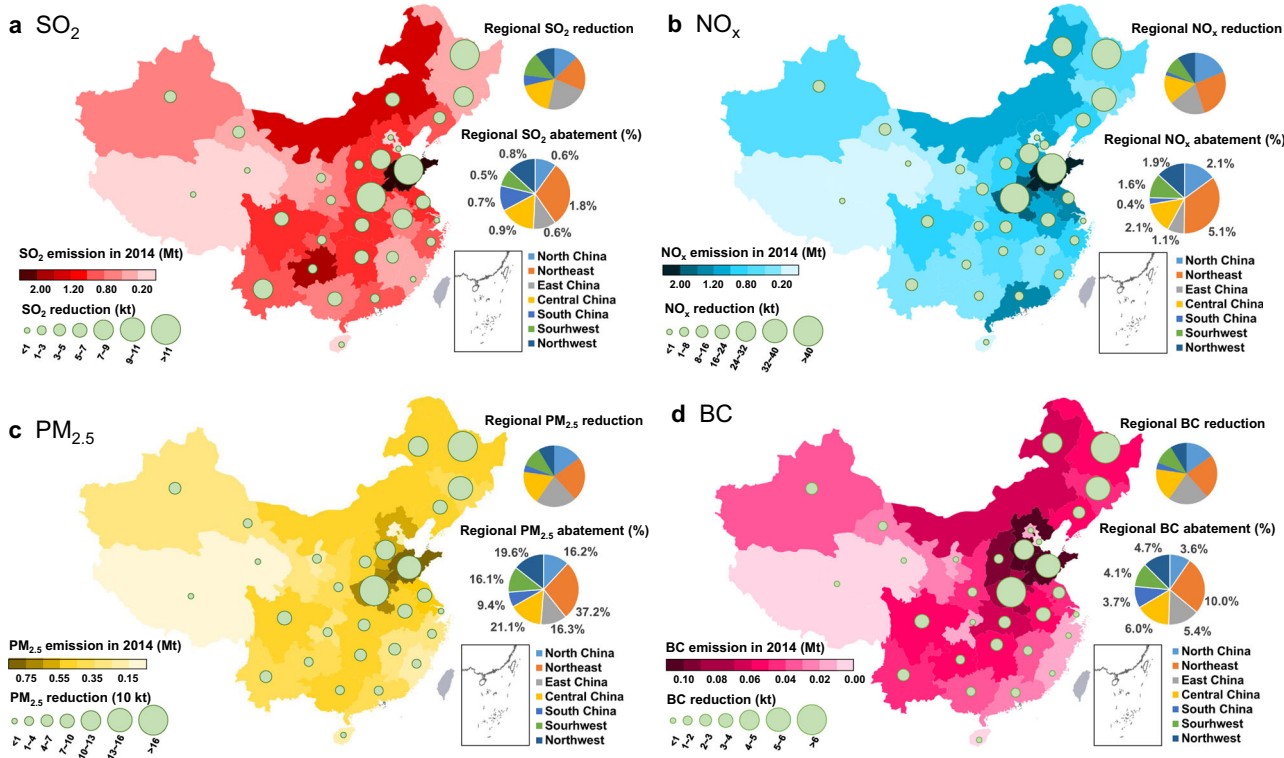

**Fig. 5 Reductions in annual air pollutant emissions (SO₂, NOₓ, PM₂.₅, and BC) achieved by nationwide deployment of BIPP systems with biochar sequestration. a** The annual SO₂ emissions reduction, the red color shading represents the levels of SO₂ emissions in 2014; **b** the annual NOₓ emissions reduction, the blue color shading indicates the levels of NOₓ emissions in 2014; **c** the annual PM₂.₅ emissions reduction, the yellow color shading shows the levels of PM₂.₅ emissions in 2014; **d** the annual BC emissions reduction, the pink color shading represents the levels of BC emissions in 2014. The green round labels in **a–d** indicate total emissions reductions by province resulting from three sources: BIPP substitution for other energy uses, avoided domestic biomass burning (DBB), and avoided open biomass burning (OBB). The regional divisions in **a–d** are shown in Supplementary Fig. 9, and the regional pollutant emissions abatement (%) is defined as the regional emissions reduction divided by the regional emissions in 2014.

in BIPP systems); and pollution avoided by reduced domestic burning of biomass waste and the open burning of biomass in fields[40]. The results indicate that deployment of BIPP systems can provide for a significant reduction in China's air pollution, including decreases in emissions of SO₂, NOₓ, BC, and primary PM₂.₅. The results of the simulations indicate that nationwide deployment of BIPP could contribute to a reduction in annual emissions of SO₂ by 0.14 Mt (0.72%), NOₓ by 0.41 Mt (1.85%), BC by 0.07 Mt (5.33%), and PM₂.₅ by 1.76 Mt equivalent to 19.32% of total national PM₂.₅ emitted in 2014[41]. In addition, deployment of BIPP could have even more pronounced effects regionally, especially in Shandong, Hebei, and Henan provinces in North and East China, which are currently suffering severe air pollution (Fig. 5). Deployment of BIPP systems in the above three provinces has the potential to reduce emissions of SO₂ (by 1.76%), NOₓ (by 5.06%), PM₂.₅ (by 37.22%), and BC (by 9.97%).

**Scenario analyses of BIPP technology deployment.** The seven scenarios of BIPP deployment are developed and can be divided into two groups: the first "Moderate development of bio-NETs" (scenarios 1–5) includes options assuming bio-NETs deployment in different sustainable biomass allocations considering China's use practices and policies for the collection and utilization of crop residues; the second "Maximum bio-NETs potentials" (scenarios 6–7) considers more ambitious options assuming optimal conditions aimed at maximizing China's contributions to climate change mitigation by exploiting the full potential of its sustainable biomass resources in biochar and biofuels (Table 2, explained in Supplementary Note 17 in detail). Note that scenarios 2–4 and

6–7 assume coordinated development of BIPP and BECCS after 2030: under these scenarios, a share of BIPP systems will be transformed into BECCS systems directly through retrofitting, changes of key reaction parameters, and CCS technology added to flue gas outlets; the remaining BIPP systems not yet transformed into BECCS systems will be coupled with CCS after 2030 (as explained in detail in "Methods"). The BIPP systems in these scenarios are based on the system analyzed above, including collection and transport of crop residues and biochar application to fields.

Compared with a Business-as-Usual (BAU) case (explained in Supplementary Note 17), the results from the seven GHG emissions reduction scenarios are shown in Fig. 6. In the "Moderate development of bio-NETs" scenarios (1–5), it is clear that the cumulative GHG emissions reduction by 2030 increases with the growth in BIPP deployment between 2020 and 2030, which can make an important contribution to the decarbonization of energy use and to a slower accumulation of CO₂ in the atmosphere up to 2030[31,42]. In the mid-term (2030–2050), although BECCS is believed to have considerably larger negative CO₂ emissions[12,43], the results show that larger early deployment of BIPP including biochar production coordinated with some BECCS (scenario 2–4) can have larger cumulative GHG emissions reduction than gradual deployment of BECCS from 2030 (scenario 1). For a mid-term perspective, the initial highest biomass input in BIPP coordinated with further BECCS deployment (scenario 4) can reach up to a 8620 Mt CO₂-eq reduction by the end of 2050, 4555 Mt and 4590 Mt CO₂-eq greater, respectively, than the BECCS-only or BIPP-only

**Table 2 Annual China's available biomass feedstock allocation (explained in Supplementary Note 17) for bio-NETs (BIPP and/or BECCS) during 2020–2050.**

| Groups | Scenarios | 2020–2030 | 2030–2050 |
|---|---|---|---|
| Moderate development of bio-NETs | Scenario 1 (S1) | – | BECCS only<br>crop residues used in BECCS increase from 0 to 80% by 2050 |
| | Scenario 2 (S2) | BIPP<br>11% of crop residues are used in BIPP for biochar sequestration and pyrolysis gas use | BIPP + BECCS<br>11% of crop residues, used in BIPP for biochar sequestration and pyrolysis gas use, transitioned to BECCS gradually by transformation of BIPP to BECCS; crop residues in BECCS increase from 0 to 69% by 2050 |
| | Scenario 3 (S3, base case) | BIPP<br>33% of crop residues are used in BIPP for biochar sequestration and pyrolysis gas use | BIPP + BECCS<br>33% of crop residues, used in BIPP for biochar sequestration and pyrolysis gas use, transitioned to BECCS gradually by transformation of BIPP to BECCS; crop residues in BECCS increase from 0 to 47% by 2050 |
| | Scenario 4 (S4) | BIPP<br>73% of crop residues are used in BIPP for biochar sequestration and pyrolysis gas use | BIPP + BECCS<br>73% of crop residues, used in BIPP for biochar sequestration and pyrolysis gas use, transitioned to BECCS gradually by transformation of BIPP to BECCS; crop residues in BECCS increase from 0 to 7% by 2050 gradually |
| | Scenario 5 (S5) | BIPP<br>73% of crop residues are used in BIPP for biochar sequestration and pyrolysis gas use | BIPP only<br>73% of crop residues are used in BIPP for biochar sequestration and pyrolysis gas use; crop residues in biochar increase from 0 to 7% by 2050 gradually |
| Maximum bio-NETs potentials | Scenario 6 (S6) | BIPP<br>100% of crop residues are used in BIPP for biochar sequestration and pyrolysis gas use | BIPP + BECCS<br>100% of crop residues, used in BIPP for biochar sequestration and pyrolysis gas use, transitioned to BECCS gradually by transformation of BIPP to BECCS |
| | Scenario 7 (S7) | BIPP<br>100% of biomass including crop residues, Miscanthus and forest residues are used in BIPP biochar sequestration and pyrolysis gas use | BIPP + BECCS<br>100% of biomass including crop residues, Miscanthus and forest residues, used in BIPP for biochar sequestration and pyrolysis gas use, transitioned to BECCS gradually by transformation of BIPP to BECCS |

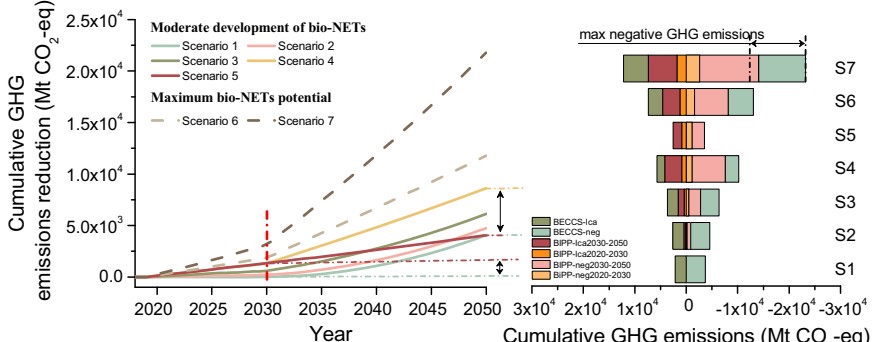

**Fig. 6 Cumulative GHG emissions reduction during 2020–2050 compared with BAU and cumulative GHG emissions.** The net life-cycle GHG emissions are studied taking account of carbon fixation by biochar/carbon storage by BECCS, life-cycle GHG emissions for a BIPP/BECCS system and biochar soil effect of biochar sequestration (see "Method", Eq. (2)). The GHG emissions reduction is attributed to biochar production in the studied pyrolysis (BIPP) systems or/and BECCS (including fossil fuel offset and net life-cycle GHG emissions). Results are shown for seven scenarios, in which solid lines indicate the scenarios under "Moderate development of bio-NETs", dashed lines under "Maximum bio-NETs potentials".

scenarios with biochar sequestration during 2020–2050 (scenario 1 and 5). To put this in a worldwide context, the BECCS global emissions reduction target required to meet the Paris Agreement goal of limiting the increase in global temperatures to no more than 1.5 °C by 2050 was pegged at 28–65 Gt $CO_2$-eq[31]. China by itself can achieve around 13–31% of the global removal goal by

focusing strongly between now and 2050 on deployment of BIPP systems including biochar production through the detailed but conservative analysis employing only agricultural, forest residues, and energy plants (explained in Supplementary Note 21). In addition, given that the biochar soil amendment is beneficial to biomass cultivation (see in Supplementary Note 13), the

near-term BIPP deployment with biochar sequestration in soils has the potential to make an important contribution to global negative emissions in the second half of this century[31] (shown in Supplementary Note 17).

The Maximum scenarios (defined above) represent more ambitious options for GHG emissions reduction using BIPP with biochar production and sequestration. The results vary depending on available biomass feedstock. The scenario using 100% of agricultural residues (scenario 6), and that which includes also energy crops and forest residues (scenario 7) could provide for annual reduction potentials of 1860 Mt or 3156 Mt $CO_2$-eq by 2030, respectively, in comparison to BAU. Moreover, the GHG emissions reduction potential for scenario 7 could reach up to 21,803 Mt $CO_2$-eq by 2050, in which the negative emissions inferred from the net life-cycle analysis would be 11,484 Mt $CO_2$-eq. Thus, in the national context, the options under the Moderate or Maximum scenarios for BIPP deployment before 2030 could contribute to a 198–3156 Mt $CO_2$-eq GHG reductions in comparison to BAU. These options can reduce carbon emissions per unit of 2005 GDP by 2–61%, which could approach or even almost achieve the goal for a 1.5 °C warming limit advanced in the emissions target proposed in China's NDC for the Paris Agreement.

This course of action has three main advantages: first, it can reduce the uncertainty and the risks associated with deployment of BECCS while mitigating any trend to overshoot goals to limit carbon in the near term; second, it takes advantage of the synergy between BIPP-based biochar/biofuel production and BECCS, promoting a larger GHG emissions reduction and even negative emissions in the mid-term; third, the deployment of BIPP will provide incentives for the collection and distribution of agricultural feedstocks for expanded BECCS applications over time.

## Discussion

The analysis indicates that it is important to have a comprehensive assessment (including technical, economic, and environmental analysis) of BIPP technology with biochar production, and to extend study of the operation of a single plant to a national scale so as to explore the sustainable pathways toward meeting GHG emissions reduction goals. Every stage of the analysis plays a critical role in assessing China's BIPP and biochar benefits in the near term. As modeled here, with trade-offs among technology, economy, and environment, a BIPP plant coupled with soil application of resulting biochar and the use of resulting pyrolysis gas and bio-oil could contribute to negative carbon emissions and reductions in air pollutant emissions while at the same time generating profit, a win–win scenario. It is suggested that to achieve economic viability and environmental sustainability, the BIPP reactors should be operated at optimal temperatures of between 550 and 650 °C, which can provide products that can be beneficial both to industry and to society as a whole. Furthermore, to ensure orderly development of the industry, a system to support production of biomass feedstocks should be initiated and implemented. This support structure should include integrated collection–transportation–pretreatment facilities in rural areas, increasing bio-diversity of feedstock resources through inclusion of energy crops and forest residues, and finally developing material storage facilities to guarantee availability of biomass feedstock during all seasons. It cannot be overstated that policy incentives and farmer awareness of opportunities for collection of crop residues will form an essential basis to guarantee stable and sufficient biomass feedstock availability for processing facilities, ultimately not only for BIPP facilities but also for BECCS when it becomes viable.

In addition, the national deployment strategy for BIPP should be focused first on areas with both abundant biomass resources and urgent needs to curb GHG emissions and air pollutant emissions, for example, Shandong, Hebei, and Henan provinces. These areas could represent cradles for demonstration of the proposed concept. Meanwhile, taking account of the poor economic conditions of areas such as Henan due to relatively expensive feedstock prices, it is advised that BIPP systems in such high-cost areas be subsidized. Second, provinces with abundant biomass resources and stronger economic conditions for deployment of BIPP technology should be incentivized to deliver the pyrolysis gas-fired electricity from BIPP systems to areas which have severe air pollution but insufficient supplies of biomass feedstock, e.g., Shanxi, Guizhou, and Inner Mongolia. The application of BIPP systems with biochar amendment can take full advantage of local agricultural biomass, imparting at the same time benefits in GHG emissions mitigation and the near-term abatement of air pollution.

Allocation of crop residues for BIPP production of biochar and pyrolysis gas, as well as for deployment of BECCS over the intermediate time period, were also analyzed and demonstrated. Compared with sole dependence on BECCS in the future, immediate implementation of large-scale BIPP facilities with future deployment of BECCS not only could decrease the GHG emissions intensity of energy in the near term, but could result also in greater cumulative reduction of GHG emissions by 2050 as compared with waiting for BECCS to become widely viable. Additionally, deployment of this strategy can contribute effectively to long-term net negative GHG removal by 2100, since it will result in higher annual negative emissions for bio-NETs by 2050. Furthermore, it is shown that China, through applications of BIPP biochar and pyrolysis gas production as BECCS matures, could even under a moderate deployment achieve potentially one-third of the GHG reduction identified in the IPCC's global NETs assessment.

It should be emphasized that the above life-cycle analysis did not consider the consumption of water during biochar deployment. The effect of water consumption during biochar production will be a topic for detailed analysis in our future work, combining potential water saving effects on soil in conjunction with biochar sequestration, offering important advantages as compared with water-intensive BECCS. These studies will include a comprehensive evaluation of water constraints for large-scale biochar production and sequestration, allowing recalculation of the GHG footprint of biomass pyrolysis poly-generation systems, and provide guidelines for policy incentives directed at greater biochar production and greater use for it in the future as a medium for carbon storage.

Finally, it is important to stress that despite the overwhelming evidence for the potential benefits of biochar on soil quality or crop productivity, such effects were not considered in the analysis of the BIPP systems, due to the complexity and specificity of such effects.

## Methods

**System model**. The moving-bed BIPP system was simulated using Aspen Plus software verifying results with data provided from experimental[20] and demonstration plants in China[26]. The main pyrolysis product yields were simulated with a model based on mass and energy flows and each reaction step was tuned using available experimental results. The detailed information on model assumptions, model parameters, inputs and outputs of materials, and energy and water for the BIPP system is summarized in Supplementary Tables 1–6.

**Economic benefit and cost**. In this study, given the immaturity of the biochar market and the lack of policies supporting biochar production and sequestration in soils in China, this study assumes that the biochar used in cultivation for improving soil is transported from BIPP systems to the field directly. There is thus no profit

from biochar for the pyrolysis system considered here in contrast to the assumption made in previous research[28]. The revenues are instead derived mainly from co-production of pyrolysis gas used for household heating and electricity production and bio-oil (both heavy and light oil) employed as a chemical feedstock. The prices for all products are shown in Supplementary Table 9, confirmed according to a detailed feasibility report performed on the existing Chinese demonstration plant[26] (noting that the technology used in the plant has received the "Blue Sky Award" from the United Nations Industrial Development Organization in 2014). However, the price of pyrolysis gas is influenced by its quality, based on the market price of syngas. It is assumed that when the lower heating value of pyrolysis gas is under 10 MJ m$^{-3}$, gas can be sold at $4.54 \times 10^{-2}$ USD Nm$^{-3}$, and otherwise at $6.81 \times 10^{-2}$ USD Nm$^{-3}$. In addition, it is assumed in this study that 20% of BIPP pyrolysis gas is used for household heating while the remainder is used to produce electricity.

To have an integrated economic assessment, this study lists an inventory for the cost of the whole process, including the construction and O&M phases of related facilities. The construction data were taken originally from the "Feasibility Report for Moving-bed Pyrolysis System"[26]. However, given the difference in scale between the demonstration plant in the feasibility report and the one analyzed in the simulations, the construction cost was estimated by the Aspen Process Economic Analyzer (APEA), except for equipment that cannot be estimated from APEA. Cost of the latter was calculated from the equation:[44]

$$C_1 = C_0 \left( \frac{S_1}{S_0} \right)^n \qquad (1)$$

where $C_1$ is the newly calculated equipment cost with $S_1$ capacity, $C_0$ is the initial equipment cost with $S_0$ capacity, and $n$ is the scaling factor, typically 0.6, the reasons and details are provided in 2011 ethanol report from National Renewable Energy Laboratory[1]. In this study, the initial equipment cost refers to the cost in the feasibility report.

The O&M component includes the collection cost, the loan interest connected with the initial investment, and the net power generation and water consumption, which can be calculated in the Aspen simulation. Supplementary Tables 9–11 show the detailed economic evaluation parameters and financial indicators of this part of the analysis.

**Net life-cycle GHG emissions for the system.** In this study, the net life-cycle GHG emissions ($E_{net}$) were calculated accounting for three components: the life-cycle GHG emissions associated with crop cultivation, transportation and the production processes of material inputs required by whole BIPP system ($E_{BIPP}$), the fixed $CO_2$ (carbon fixation) of biochar sequestration ($CO_2$ $_{fixed}$), and the biochar soil effect of biochar sequestration ($E_{soil}$), as shown in Supplementary Fig. 5:

$$E_{net} = E_{BIPP} - \left( E_{soil} + CO_{2\,fixed} \right) \qquad (2)$$

The biochar soil effect results from the changes in physical and chemical properties of soil consequent to the addition. Biochar additions are believed to be beneficial in terms of reductions in soil $N_2O$ and $CH_4$ emissions. Noting that there is controversy about the nature of the $CH_4$ emissions reduction[44], however, this study assumes that sequestration of biochar into soil has favorable impact only for $N_2O$ emissions. There are also significant uncertainties associated with the reductions in soil $N_2O$ emissions associated with various rates and types of biochar application (e.g., biochar quantity and applied depth) and soil condition (e.g., soil type)[44]. Thus, the calculation method is taken directly from previous research[44] (described in Supplementary Note 13).

The life-cycle GHG emissions have been assessed for a specific BIPP system based on modeling results for operational emissions and the literature[45] defining GHG emission coefficients for material utilization in every sub-process. Detailed data are summarized in Supplementary Table 14. Furthermore, the $CO_2$ fixation ($CO_2$ $_{fixed}$) in units of t per individual BIPP plant) in the biochar is calculated according to:

$$CO_{2\,fixed} = P_{BC} \times C_{BC} \times S_{BC} \times CF \qquad (3)$$

where $P_{BC}$ indicates the biochar yield from the pyrolysis system (t per plant), $C_{BC}$ represents the fixed carbon content of biochar (t fixed carbon per t biochar), $S_{BC}$ measures the stability rate of carbon in soil (%), and CF defines the C-$CO_2$ conversion factor, taken here as 3.67. Thus, the net life-cycle GHG emissions potential is evaluated for the BIPP plant selected for the detailed study reported in this paper.

**National GHG emissions reduction potential.** Calculation of the national GHG emissions reduction ($E_{red}$) is based on the total GHG emissions reduction for one BIPP system, which is calculated by two components: the net life-cycle GHG emissions and the substitution of fossil fuels ($E_{offset}$):

$$E_{red} = E_{offset} - E_{net} \qquad (4)$$

The substitution of fossil fuels refers to GHG emissions saved from pyrolytic products (i.e., pyrolysis gas, electricity, and bio-oil) as compared with conventional products produced from fossil fuel sources and processes. This study considered that pyrolysis gas and pyrolysis gas-fired electricity from BIPP system can

substitute for coke oven gas and for electricity generated from coal-fired power plants. The product bio-oil can be used as an alternative to coal tar developed as a by-product of coke production. Thus, the substitution of fossil fuels resulting from BIPP can be calculated as:

$$E_{offset} = P_{gas} \times EF_{gas} + P_{elc} \times EF_{elc} + P_{oil} \times EF_{oil} \qquad (5)$$

where $P_{gas}$, $P_{elc}$, and $P_{oil}$ indicate, respectively, the yields of pyrolysis gas, electricity, and bio-oil from the corresponding BIPP deployment (in Nm$^3$, kWh, and t, respectively); $EF_{gas}$ represents the GHG emissions intensity for coke oven gas production (g $CO_2$-eq Nm$^{-3}$); $EF_{elc}$ represents the GHG emissions intensity of the electricity generated by standard coal-fired power plants (g $CO_2$-eq kWh$^{-1}$); and $EF_{oil}$ represents the GHG emissions intensity of coal tar from the standard coke wet quenching process (g $CO_2$-eq t$^{-1}$). Detailed data are shown in Supplementary Table 21.

Thus, given that the total GHG emissions are sensitive to biomass type, working parameters, geographical characteristics, and other factors, in order to make a reasonably accurate assessment of the effects on national GHG emissions, it is assumed that every system nationwide runs under optimum operating conditions in terms of economic and environmental effects, while using provincial-level data and experimental results to evaluate the differentiated influences from biomass types and geographical conditions. The detailed explanation can be seen in Supplementary Note 14 and Supplementary Fig. 6.

**Scenarios.** Based on the current state of China's utilization of crop residues and prospects for exploitation of bioenergy NETs (BIPP and BECCS), two groups of scenarios ("Moderate" and "Maximum") based on sustainable biomass availability in China were assumed. As defined scenario-by-scenario in Table 2, the first group (moderate scenarios 1–5) assumes realistic levels of biomass availability between 2020 and 2050 and is labeled "Moderate development of bio-NETs"; the second group (maximum scenarios 6–7) assumes very high levels of biomass availability and is labeled "Maximum bio-NETs potentials".

In scenarios 1, 2, 3, 4, 6, and 7, based on current state-of-the-art research[12,13] and in accordance with scenarios analyzed in IPCC reports summarized by Kemper[7], it is assumed that BECCS technologies can be deployed on a large scale starting gradually in 2030. On the other hand, biochar production using BIPP technology is arguably readily available for deployment now and likely to be promoted in the near future in China. For these reasons, the scenarios cover two sequential time periods, including 2020–2030 when only BIPP will be available for deployment and 2030–2050 when both BIPP and BECCS will be available. In addition, to explore the situation without CCS development (in case of its failure to become cost-effective), scenario 5 only focuses on BIPP without CCS from 2020 to 2050. Analysis of these scenarios enables us to identify preferable courses of action for the near-term deployment of biochar technologies to maximize GHG emissions reduction, providing critical and timely inputs for policy makers designing strategies and policies affecting bioenergy NETs.

**Sustainability criteria.** In consideration of multiple use paths for crop residues in China, the criteria for sustainable biochar production in the baseline scenario (scenario 3 in Table 2) requires that the biomass feedstock used in pyrolysis accounts for 33% of collected crop residues over the period 2020–2030, a percentage corresponding to the residues currently used as fuel (mainly for household cooking and heating) or burned directly in the field. Thus, the national pyrolysis application of the baseline scenario does not require a change in the amount of total biomass utilization, with residues still producing energy supply (i.e., the electricity and pyrolysis gas) from the BIPP systems. It is further assumed in additional moderate scenarios (numbers 1, 2, 4) that the BIPP feedstock for 2020–2030 is 0%, 11%, and 73% of the total current availability of crop residues, respectively, and by 2050 it could achieve 80% by retaining the minimum of 20% required for livestock feed (rationales for these percentages are explained in Supplementary Note 17).

For scenarios 2–4 and 6–7 in the period 2030–2050, this study assumed that BECCS is by then mature and can be developed gradually in coordination with BIPP. Biomass resources will be used both in newly built BECCS plants and retrofitted BIPP plants.

First, it is assumed that BECCS is deployed from 2030 onward and the biomass used in BECCS increases steadily through gradual deployment, rising to 69% (in scenario 2), 47% (in scenario 3), and 7% (in scenario 4) by 2050, respectively. In addition, because of the superior carbon sequestration potential of BECCS, it is assumed that the biomass used in BIPP systems will then be gradually transitioned through retrofitting of plants and changes in key reaction parameters, at annual rates of 0.74%, 2.7%, and 13.0%, respectively. The calculation equation is as follows:

$$R_i = \sqrt[20]{\frac{Per_{target}}{Per_i}} - 1 \qquad (6)$$

where $R_i$ indicates the annual rate of increase of biomass used in BECCS for scenario $i$th (scenarios 2–4), in which the biomass increases from the BIPP system transformation; $Per_{target}$ and $Per_i$ are the percentages of crop residues in BECCS in 2050 and 2030, respectively, and, in this study, it is assumed that the targeted

percentage would be 80% of residues in BECCS. The percentage for each scenario is explained in Supplementary Note 17.

For scenarios 6–7 exploring the maximum bio-NETs potentials, the BIPP systems would be developed by making use of all biomass (100% of crop, forest residues, and energy crops) during 2020–2030. All of the BIPP plants would then be transitioned to BECCS gradually with an annual average change rate of 5% in the period 2030–2050. In these scenarios, the BIPP systems that have not been transformed into BECCS will be coupled with CCS technology after 2030. It is assumed that the $CO_2$ produced from power generation and biomass fuel combustion is captured by monoethanolamines, compressed, and stored underground (BIPP + CCS) from 2030 onward.

**Comparison with BECCS**. To get comparable results for GHG emissions reductions and negative emissions by 2050, the same models and biomass planting and transportation sustainability criteria for BIPP and BECCS applications were applied. At the same time, it is assumed that BECCS systems retrofitted from BIPP plants have the same scale (i.e., biomass feedstock input) during the period 2030 to 2050.

**Reporting summary**. Further information on research design is available in the Nature Research Reporting Summary linked to this article.

## Data availability
The data that support other plots within this paper and other findings of this study are available on request from the corresponding author. Source data are provided with this paper.

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

## Acknowledgements

This work was supported by National Natural Science Foundation of China (No. 52076099), the Foundation of State Key Laboratory of Coal Combustion (No. FSKLCCA1902), and the Double first-class research funding of China-EU Institute for Clean and Renewable Energy (No. 3011120016). We also would like to thank members of the Harvard-China Project on Energy, Economy and Environment for useful comments and suggestions, and the Harvard Global Institute for an award to the Harvard-China Project on Energy, Economy and Environment.

## Author contributions

Q.Y., H.C, and M.B.M. designed research; Q.Y., H.Z., H.C., C.P.N., and M.B.M. performed research and manuscript preparation; Q.Y., Z.W., and H.Z. contributed the simulation for pyrolysis system by Aspen Plus; H.Z., Q.Y., P.B., O.M., and X.L. contributed the integrated analytic tools; H.Z., Q.Y., P.B., and O.M. analyzed data; and H.Z., Q.Y., P.B., F.F., O.M., F.A.A., H.Y., H.C., X.L., G.C., C.Z., C.P.N., and M.B.M. discussed the results and implications, contributing to sections of the manuscript, Methods and Supplementary Information at all stages.

## Competing interests

The authors declare no competing interests.
