## [Peer Review File · Nature Communications]

REVIEWER COMMENTS

Reviewer #1 (Remarks to the Author):

The manuscript entitled "Prospective contributions of biomass pyrolysis to China's 2050 carbon reduction and renewable energy goals" evaluated the potential of a ready-to-implement biochar technology (biomass intermediate pyrolysis poly-generation) as a negative emission technology to reduce the GHG emissions, produce biofuels and improve the soil composition. It is a complete study, focusing on technological, environmental and economic issues. In my opinion, this study is within the scope of Nature Communications. Therefore, I recommend its publication after performing minor changes in the document. Please see my comments below.

Minor comments:

- L30: Avoid the use of first-person (we, our, ...). Revise all the document.
- L228: Change the notation "E+**" to "10^" (revise SI).
- L587: "BIPP" was already defined.

Reviewer #2 (Remarks to the Author):

The paper examines the potential role of slow pyrolysis technology in a polygeneration approach to reduce greenhouse gas emissions out to 2050 in China. The paper includes an analysis of the contribution of a described biomass intermediate pyrolysis polygeneration process arrangement applied to two main agricultural crop residues and compares the greenhouse gas reduction potential of deploying the technology in the medium to long term and on a number of scales.

The paper gives a much needed example of the role biochar and slow pyrolysis technologies can play in addressing global warming challenges, and gives a fascinating insight into the massive drawdown potential of a well-engineered slow pyrolysis technology.

The paper however would benefit from revision before it can be accepted for publication, particularly in a journal of such high standing as Nature Communications.

(1) The authors attempt to cover a very large scope of analysis, however to do so they analyse a narrow set of parameters which are built on a set of key assumptions. The sensitivity of the outcomes to these key assumptions are not explored and it is therefore unclear the range of values which may be reflective of real technology potential. For example, would the benefit of biochar generation be completely outweighed if its soil carbon stability (an important and lesser known/quantified value) was reduced by a further 10%? What if electricity/heat generation in provinces was not replacing fossil fuel alternatives but in fact other non-emitting sources such as solar or wind? Is the cost of bio-oil realistic considering it would require further processing and have different characteristics in a slow pyrolysis process compared to fast pyrolysis (which is it believed is where the cost values were taken from)? 20% of syngas assumed to remain as gas for households while electricity was the balance – what if this was 40%, or 80%? Why was 20% chosen for this? If the authors were able to conduct a simple sensitivity analysis around perhaps one or two scenarios for a few key assumptions the potential of the technology could be demonstrated.

(2) The key parameters of BECCS scenarios were not made clear in the document. Nowhere did the authors state the type of technology which would be utilised in this case (biomass combustion followed by CO₂ capture in amines with compression and underground CO₂ storage?). It was also unclear whether the 'transition' to BECCS meant a full technology transition from BIPP to BECCS, meaning full infrastructure transition for technology, or if BIPP would contribute but with flue gas outlets retro-fitted with CCS technology? In fact this would be an interesting additional scenario to investigate or perhaps could be explored in the maximum potential emission reduction scenario.

(3) The BIPP process is not well described in the supplementary information regarding the ASPEN model used and results were somewhat difficult to interpret. Was the BIPP process optimised to ensure maximum energy efficiency/recovery? In particular the internal use of energy (including heat exchanger parameters) were difficult to interpret from information provided. The authors did

not attempt to compare the transformation or electric efficiency to any other published works regarding slow pyrolysis, making it difficult to gauge if their final energy efficiency values are reasonable. The ASPEN model diagram also appears to be incorrect and does not match with the discussion. For example, after exiting the pyrolysis reactor the vapour stream is water cooled, with the hot water outlet mixed with an air stream (HEATRECO) and then referred to as 'dry -air', the heat of which is claimed for drying. If this is an indirect contact heat exchanger this would be a very inefficient way to dry incoming biomass, but direct drying wouldn't be possible either since introducing moisture in the dryer would not be desirable. The heat flows are also shown seemingly in the wrong direction in Figure S1, pointing towards both HEAT-WST and HEATSUPP instead of away. In Section 1.2.3 (sup) SEC-EXCH is said to further cool pyrolysis vapours using 'cold water', however this water is directed from VAT-COOL (hard to read label in figure provided) and is labelled as hot water being the inlet to SEC-EXCH, cold water being the outlet which would suggest the vapour mix temperature would increase, making it a heater not cooler. AIR-PRH is also unclear showing two air inlets and outlets of unknown temperatures and no way of knowing which way heat was transferred in this unit operation. A property table for an example process basis (1tph biochar production for example) including streams flows and properties (including temperatures) would make the ASPEN model more accessible and able to be interpreted. As it stands it is difficult to judge whether the BIPP process is well designed or optimised. For example, why did the authors decide to burn biomass to make heat? Would it not be more efficient to use internally generated syngas in a burner to achieve a higher heat transfer efficiency? The use of cooling water is also not well justified. Air cooling would negate the need for water treatment, discussed by the authors to have significant greenhouse gas implications for the life cycle analysis. Of course air would have a reduced heat transfer rate compared to water, however the authors did not discuss or describe whether this benefit is balanced by the need to process water for use in water cooling. Did the authors consider recycling of water within the plant? On p13 line 249 30% of GHG emissions attributable to the plant are from waste water treatment – but this should be a very small amount of make-up water presumably if it is able to be recovered internally. The inlet moisture of the feedstock is also a major determinant for the overall energy balance of a slow pyrolysis process since it demands high energy for drying, how was the feedstock moisture determined/decided? This needs better discussion/acknowledgement in the paper and may be an important overarching assumption to do sensitivity analysis around in terms of process modelling outcomes.

Other specific comments include:

MAIN PAPER

Line 26: Deployment of low or zero emission technologies, rather than 'deployment of emission technologies'?

Line 183 and 184: Energy efficiency is said to decline gradually with the increase in pyrolysis temperature, however Fig. 1 shows 'electric power efficiency' which actually increases with temperature (along with increase in syngas generation, it is guessed). Presumably the authors are talking about the energy efficiency defined in Table S6 in these lines, however this is not clear. In fact it is not clear why the electric efficiency is displayed in Fig 1 instead of energy efficiency (since it is not discussed in the text) nor how electric efficiency value is calculated in the work. Electrical energy output divided by total input feedstock LHV (including biomass combusted??)?

Figure 1: It would be useful on Fig 1 a) to also show biochar generation rate to explain the change in cooling water usage as a trade-off between reaction temperature and amount of biochar produced. It is also unclear why the authors considered such low temperatures as 250°C here where the biomass would not be pyrolysed at all but rather torrefied, and certainly not stabilised at this stage. It is strange the authors chose this parameter (temperature) to investigate when there are many other process variables which would be more likely to change (including feedstock inlet moisture for example).

Line 272-274: Authors suggest it is possible to do a like-for-like replacement of fossil fuels with pyrolysis products, but do not seem to directly compare the heating value of the produced products with expected replacements. It is very unlikely that the syngas would be usable as a direct natural gas replacement, and would likely have a much lower heating value. Did the authors account for this? The method is not clearly described in this case.

Line 361: It is assumed that the BIPP process will drastically reduce particulates and other environmental pollutants, however the process outline did not show or consider any flue gas treatments, particularly for the combusted biomass stream used to generate heat for pyrolysis. Did the authors simply assume BIPP would result in no particulates, or was there a regulated standard applied to the BIPP process flue streams? NO_x may also be generated during pyrolysis when done

under nitrogen as the inert atmosphere, this did not appear to be accounted for or discussed in the work.

Figure 5: The grey bars are very difficult to read and interpret on the figure.

Line 467/468: Here the authors claim that large capacity BIPP plants can result in additional positive environmental benefits but did not explain in comparison to what or via what mechanism? In comparison to small plants? Is there a limit on the size of BIPP plants considered in the study? There will be a certain throughput for which a fully engineered process including auxiliaries would become completely impractical, this wasn't fully disclosed in the work.

Line 516/517: BECCS was said to be highly water intensive (due to amine capture technology?) but no information on the technology was provided to determine or compare the water use outlined here (cooling water usage was not given here either).

Eqn (3): The authors should details the units for each of the variables used in equations. It is presumed CO₂ fixed is in kg while PBC is also kg, CBC and SBC are percentages and CF is MW of CO₂ divided by MW of C for a 1:1 molar conversion?

Line 658: It is unclear why 0%, 11% and 73% were chosen as scenario basis, and the 'gradual' rate of increase is also seemingly arbitrarily used? Justify further why these rates were used.

Line 683: Why 5%?

SUPPLEMENTARY DOCUMENT

Line 133: Fig S1 said to show the distribution of gaseous products of rapeseed straw. In fact Fig S1 is the ASPEN model and the gaseous product distribution cannot be found in figures presented, perhaps the authors mean Table S5?

Line 216: The value of bio-oils, heavy and light, is questionable from slow pyrolysis processes. The authors should justify their use of these values in more detail.

Line 209-217: It is a strange assumption that the biochar has no value to the process. Considering the value of bio-oils is also variable and not fully established it surely will call into question why slow pyrolysis as considered at all. It is understood that the focus is mainly to reduce greenhouse gas emissions but without a stronger economic driver for making biochar (compared to just doing fast pyrolysis for example and preferencing bio-oil formation) it seems unlikely that BIPP technology would be competitive. Surely the authors should also consider a case where biochar does have an economic value to demonstrate broader application and competitive uptake?

Line 24 and 235: Dismisses BIPP [plants being built in locations where available biomass is abundant and easily supplied. This is an example of where a large assumption is applied in order to allow for a broad overarching model to be used. Perhaps this would be true in the case of low biomass utilisation scenarios, but would be highly unlikely in the case of the 80 or 100% usage case such as scenario 7?

Line 335: Authors discuss the use of diesel vehicles. While this might be true now by 2050 surely this would be hydrogen or electric in nature? It would be interesting to have more detail in here including bringing in other predictions made around hydrogen usage in agriculture for example.

Line 355-357: Not enough information is provided about wastewater treatment. For example and amount of methane production annually is given but the amount of treated waste water generated by this is not supplied for reference.

Line 360: 'in used' should be deleted.

Line 477-478: Should justify the assumption that pyrolysis technology (BIPP specifically) will allow removal of particulates mercury etc. When combusting biomass for heat inputs particulates will certainly be generated and must be treated. How can this be controlled better than existing plants also using combustion?

Table S5: Would be useful to have a standard deviation or idea of error in experimental results as this is known to have some variance and could then confirm the simulation results are within the range of experimental error.

Congratulations on an important contribution to the field of negative emission technology development and uptake.

Dr Jessica Allen
University of Newcastle, Australia

Point-by-Point Response to the Reviewer's Comments

REVIEWER COMMENTS

Reviewer #1 (Remarks to the Author):

The manuscript entitled “Prospective contributions of biomass pyrolysis to China’s 2050 carbon reduction and renewable energy goals” evaluated the potential of a ready-to-implement biochar technology (biomass intermediate pyrolysis poly-generation) as a negative emission technology to reduce the GHG emissions, produce biofuels and improve the soil composition. It is a complete study, focusing on technological, environmental and economic issues. In my opinion, this study is within the scope of Nature Communications. Therefore, I recommend its publication after performing minor changes in the document. Please see my comments below.

Response:

Thank you for your positive evaluation for the manuscript. All your suggestions are of great value and guiding significance for the improvements of our research work. According to your comments, we amended the relevant part in the manuscript and Supplemental Information carefully, and the detailed responses for these comments are as follows.

Minor comments:

- L30: Avoid the use of first-person (we, our, ...). Revise all the document.

Response:

We appreciate the reviewer’s comments very much. We have thoroughly checked our manuscript and Supplemental Information, and amend this problem. The following actions are examples of our changes.

Action (examples):

In **abstract** section, page 1 line 31-34, “we focus here on developing a ready-to-implement biomass intermediate pyrolysis poly-generation (BIPP)...” has been changed into “this study focuses on developing a ready-to-implement biomass intermediate pyrolysis poly-generation (BIPP) ...”

In **manuscript** section, page 4 line 74-75, “In this paper, we focus on China and a state-of-the-art biomass intermediate pyrolysis poly-generation (BIPP) system...” has been changed into “This paper focuses on China and a state-of-the-art biomass intermediate pyrolysis poly-generation (BIPP) system...”

In **Supplementary Information** section, page 21 line 658-660, “Therefore, we assumed conservatively that China’s agricultural resources and residue utilization would remain at the same level as the present situation in the first half of the century.” has been changed into “this study assumed conservatively that China’s agricultural resources and residues utilization would remain at the same level as the present situation in the first half of the century.”

- L228: Change the notation “E+**” to “10^” (revise SI).

Response:

Thanks for your kind advice. We have changed the notation “E+**” into “10^” throughout the manuscript and Supplemental Information.

- L587: “BIPP” was already defined.

Response:

Thank you for your careful review and kind suggestion. We have revised it.

Action:

In **Method** section, page 34 line 590-592, “The life-cycle GHG emissions have been assessed for a specific poly-generation pyrolysis (BIPP) system based on modeling results for operational emissions and the literature⁴⁴ defining GHG emissions coefficients for material utilization in every sub-process.” has been changed into “The life-cycle GHG emissions have been assessed for a specific BIPP system based on modeling results for operational emissions and the literature⁴⁴ defining GHG emissions coefficients for material utilization in every sub-process.”

Reviewer #2 (Remarks to the Author):

The paper examines the potential role of slow pyrolysis technology in a polygeneration approach to reduce greenhouse gas emissions out to 2050 in China. The paper includes an analysis of the contribution of a described biomass intermediate pyrolysis polygeneration process arrangement applied to two main agricultural crop residues and compares the greenhouse gas reduction potential of deploying the technology in the medium to long term and on a number of scales.

The paper gives a much needed example of the role biochar and slow pyrolysis technologies can play in addressing global warming challenges, and gives a fascinating insight into the massive drawdown potential of a well-engineered slow pyrolysis technology.

The paper however would benefit from revision before it can be accepted for publication, particularly in a journal of such high standing as Nature Communications.

Response:

Thank you for your positive evaluation for the manuscript. All your comments and suggestions are of great value and guiding significance for the improvements of our research work. According to your comments, we amended the relevant part in the manuscript and Supplemental Information carefully, and the detailed responses for these comments are as follows.

(1) The authors attempt to cover a very large scope of analysis, however to do so they analyse a narrow set of parameters which are built on a set of key assumptions. The sensitivity of the outcomes to these key assumptions are not explored and it is therefore unclear the range of values which may be reflective of real technology potential. For example, would the benefit of biochar generation be completely outweighed if its soil carbon stability (an important and lesser known/quantified value) was reduced by a further 10%? What if electricity/heat generation in provinces was not replacing fossil fuel alternatives but in fact other non-emitting sources such as solar or wind? Is the cost of bio-oil realistic considering it would require further processing

and have different characteristics in a slow pyrolysis process compared to fast pyrolysis (which is it believed is where the cost values were taken from)? 20% of syngas assumed to remain as gas for households while electricity was the balance – what if this was 40%, or 80%? Why was 20% chosen for this? If the authors were able to conduct a simple sensitivity analysis around perhaps one or two scenarios for a few key assumptions the potential of the technology could be demonstrated.

Response:

We would like to thank the reviewer very much. In response to this comment, we have added the sensitivity analysis in Supplementary Information to explore the range of values which may be reflective of real economic, technology and GHG emissions reduction potential. Key factors, including pyrolysis gas distribution, bio-oil price, carbon price, biochar usage, biochar stability, transport models and distances, and N₂O reduction, are taken into account in sensitivity analysis. The results are shown in Fig. S10 and Fig. S11. The detailed description has been shown in the action.

For question: “20% of syngas assumed to remain as gas for households while electricity was the balance – what if this was 40%, or 80%? Why was 20% chosen for this?”, the answer is 20% is according to the feasibility report of the demonstration BIPP plant. It is determined by the gas requirements from nearby households. And we appreciate your comments and suggestions very much, in the revised Supplementary Information, we have done a sensitivity analysis on pyrolysis gas distribution.

We also would like to explain specially on the question: “What if electricity/heat generation in provinces was not replacing fossil fuel alternatives but in fact other non-emitting sources such as solar or wind?”

- A quick answer is that in carbon emission estimations by main institutes [1-3], GHG emissions offset through substitution before 2050 is usually treated as emissions reduced by replacement of fossil fuels. We follow this routine to make our results comparable.
- In view of present China's condition, while the energy proportion of coal is gradually declining, it will still account for around 50% by 2030 [4]. Renewable energy is continually perceived as the main fossil fuel alternatives before 2050. Thus, we consider BIPP system

products to replace fossil fuels (especially coal in this study) instead of renewable energy such as solar and wind energy.

- In addition, all the scenarios in our study are for situations before 2050. In a very conservative estimation, if the GHG emissions offset through substitution are zero (no fossil fuels to replace), the GHG emission reduction of BIPP in this study would be equal to its life-cycle negative emission, which will be the minimum GHG emission reduction potential. In other words, if we conduct a sensitivity analysis on GHG emission offset through substitution, the results will be in the range of BIPP's life-cycle negative emission (without GHG emission offset through substitution) and its current total GHG emissions reduction (with GHG emission offset through substitution).

Reference:

- [1] Lu X, Cao L, Wang H, et al. Gasification of coal and biomass as a net carbon-negative power source for environment-friendly electricity generation in China[J]. Proceedings of the National Academy of Sciences, 2019, 116(17): 8206-8213.
- [2] Roberts K G, Gloy B A, Joseph S, et al. Life cycle assessment of biochar systems: estimating the energetic, economic, and climate change potential[J]. Environmental Science & Technology, 2010, 44(2): 827-833.
- [3] Rogelj J, Shindell D, Jiang K, et al. Mitigation Pathways Compatible with 1.5 C in the Context of Sustainable Development. Global Warming of 1.5 C. An IPCC Special Report on the impacts of global warming of 1.5 C above pre-industrial levels and related global greenhouse gas emission pathways, in the context of strengthening the global response to the threat of climate change [J]. Sustainable Development, and Efforts to Eradicate Poverty, Geneva, 2018.
- [4] China Energy Outlook 2030. (China Energy Research Society, 2016).

Corresponding change (in Supplemental Information):

6. Sensitivity analysis

6.1 Sensitivity analysis of the economic feasibility of a demonstration BIPP system

The different scenarios used in sensitivity analysis of economic feasibility of the different demonstration BIPP plant scenarios, due to the change of key factors, including pyrolysis gas distribution, bio-oil price, carbon price and biochar usage, are presented in Table S27. Key assumptions for these scenarios are discussed below, and the results are shown in Fig. S10.

The results indicate that the NPV of a demonstration BIPP system can vary widely under different scenarios (from -1.70×10^7 to 1.66×10^9 USD; the base case is 2.44×10^6 USD) under the different scenarios. Especially under the 1.5/2 °C temperature control target (scenario C), the carbon price could reach as high as 6050 USD₂₀₁₀/t³⁷ and then the NPV would achieve 1.66×10^9 USD. This high profit potential would help the system dramatically enhance economic competitiveness in the energy market. The bio-oil price will also change the NPV significantly. For other scenarios regarding changes of pyrolysis gas use distribution and biochar usage, the NPV does not change dramatically. It is concluded that the economic feasibility is more sensitive to the prices of products, than to pyrolysis gas use distribution and biochar usage. Therefore, a stable and mature market is a prerequisite to ensure the stable development and application of the BIPP system.

Scenario A. Pyrolysis gas distribution

In the base case simulation, according to the feasibility report of the demonstration BIPP plant, 20% of the pyrolysis gas produced by the BIPP plant is provided for household heating and 80% is used for power generation. In the sensitivity analysis, two extreme cases were simulated: 0% pyrolysis gas for household heating + 100% pyrolysis gas for power generation, and 80% for household heating + 20% for power generation (shown in Table S27). Note that reserving 20% of the pyrolysis gas for power generation limit is determined by the auxiliary power consumption of the demonstration BIPP plant.

Scenario B. Price of bio-oil

Bio-oil is an important co-product from the BIPP system. In the base case, the price of bio-oils (heavy and light oil) is the average value in the *Purchase and Sale Agreement of the Ezhou BIPP Plant* that this particular BIPP company sets with other companies. However, it is believed that the price of bio-oils could vary with the change of quality under different reaction

condition, and also could vary over time in the market. Considering the complexity and difficulty of predicting bio-oils price, they are assumed to vary according to the fluctuation of crude oil price.

As of June 2020, the average annual price of crude oil for 2020 stood at 39.89 USD per barrel. Between 1998 and 2020, the lowest and highest crude oil price reached 12.8 (1998) and 111.63 (2012) USD per barrel, respectively⁶¹. The change range of crude oil price is thus calculated to be -67.9% - 179.8%. Accordingly, this range is applied to the bio-oil price in the sensitivity analysis (shown in Table S27).

Scenario C. Carbon price

A carbon market can be an important tool in carbon reduction. In the base case, the carbon price in the European Union Emission Trading Scheme (EU-EST) is chosen to assess the deployment of BIPP systems in a mature carbon market. In order to explore the development potential of BIPP systems in the future, a range of carbon prices under the framework of the Paris Agreement has been considered (Table S28) in the sensitivity analysis.

Scenario D. Biochar usage

The biochar sequestration in soil would result in negative carbon emissions, from which BIPP plants could benefit in the carbon trading market. 100% biochar is sequestered in soil in the base case. However, in feasibility report of the demonstration BIPP plant, biochar is also recommended to be sold as solid fuel in industrial market. The biochar usage in industrial market could bring different economic benefits. In the sensitivity analysis, two cases are analyzed to consider the potential economic benefits (NPV): 60% biochar for soil sequestration + 40% biochar selling in the industrial market; and 100% biochar selling in the industrial market (shown in Table S27).

6.2 Sensitivity analysis of the life-cycle GHG emissions for a demonstration BIPP system

In order to explore the sensitivity of the life-cycle GHG emissions of the demonstration BIPP plant, scenarios with variations of the key factors, including biochar stability, transport models

and distances, N₂O reduction and biochar usage, are analyzed and listed in Table S27. Key assumptions for these scenarios are discussed below. The results are shown in Fig. S10.

It is found that the most sensitive factors affecting the BIPP's life-cycle GHG emissions are closely related to biochar, including biochar stability (scenario E, -19.7% to 34.6% relative to the base case) and biochar usage (scenario I, 0% to 199.1% relative to the base case). Biochar stability could significantly influence the amount of carbon ultimately stored in the soil. The net life-cycle GHG emissions would increase dramatically (+199.1%) when 100% of biochar is sold for uses where the carbon is ultimately released into the atmosphere (scenario I).

Scenario E. Biochar stability

Based on results of stability experiments on biochar from different types of crop residues, an average 73% of the carbon in biochar is stable over long term is assumed in the base case. In order to have a better understanding of the potential impact of this factor on the net life-cycle GHG emissions from BIPP systems, a sensitivity analysis was conducted considering the change of biochar stability in the range of 60%-80%²⁸ (a more conservative range than that in IPCC report: 79%-91% biochar stability over a time span of 100 years⁶²).

Scenario F. Transport distance

For the base case scenario, BIPP plants are assumed to be built in locations where available biomass is abundant and easily supplied with an average transport distance of 40 km (based on the BIPP demonstration plant in case study ¹¹). However, in extreme situations assuming use of more than 80% of the China's biomass resources in BIPP systems, the transport distance would be expanded so as to satisfy the daily feedstock requirements from a certain BIPP plant. Thus, in sensitivity analysis, biomass transport distance changing from 20 km to 100 km is considered (shown in Table S27 and Table S30).

Scenario G. Transport models

Based on the current status of vehicle utilization in China, the transport model of diesel vehicle is assumed in the base case life-cycle GHG emissions assessment. It is obvious that different transport models could lead to different GHG emissions. Hydrogen and electric vehicles, low-

carbon transport options, may be feasible in the future, and thus have been considered in the sensitivity analysis. Table S29 shows the energy consumption of different transport models, and the resulting net life-cycle GHG emissions from different transport models are shown in Table S30.

Scenario H. N₂O reduction from biochar soil effect

There are significant uncertainties associated with the reduced soil N₂O emissions for various biochar application rates (e.g. biochar quantity and applied depth) and soil condition (e.g. soil type)³⁰. The annual avoided soil N₂O emissions (E_N) are calculated based on equation 10, in which the R_N (reduction factor) has been investigated in the range of 0 - 80% in previous research³⁰ (60% in the base case). Based on this range, the potential N₂O reduction from biochar soil effect is thus considered in the sensitivity analysis (Table S27).

Scenario I. Biochar usage

If biochar is not returned back to the soil (with a negative emission), it can be sold as fuels. The carbon in biochar will be released back into atmosphere (with a neutral emission). Considering the GHG emission difference between biochar sequestration in soil and use of biochar as a fuel substitute, is studied on three cases in the sensitivity analysis: 100% of biochar for sequestered in soil (base case); 60% biochar in soil sequestration + 40% sold in the industrial market; and 100% biochar selling in the industrial market (shown in Table S27).

Table S27. Scenarios for sensitivity analysis

Scenarios		
Economic feasibility		
A (Pyrolysis gas distribution)	Lowest	0% household gas + 100% power generation
	Highest	80% household gas + 20% power generation
B (Price of bio-oil)	Lowest	Price after change rate of -67.9% for base case
	Highest	Price after change rate of 179.8% for base case
C (Price of carbon price)	Lowest	Carbon price -15 USD ₂₀₁₀ /t
	Highest	Carbon price -6050 USD ₂₀₁₀ /t
D (Biochar usage)	Highest	100% biochar selling
	Lowest	40% biochar selling + 60% biochar sequestration in soil

Life-cycle GHG emissions		
E (Biochar stability)	Lowest	Biochar stability -80.0%
	Highest	Biochar stability -60.0%
F (Transport distance)	Lowest	Average distance for biomass feedstock -20 km
	Highest	Average distance for biomass feedstock -100 km
G (Transport method)	Lowest	Biomass transport by electricity vehicles
	Highest	Biomass transport by gasoline vehicles
H (N ₂ O reduction from biochar soil effect)	Lowest	Reduction factor for N ₂ O emissions (R _N) -80%
	Highest	Reduction factor for N ₂ O emissions (R _N) -0%
I (Biochar usage)	Highest	100% biochar selling
	Lowest	40% biochar selling + 60% biochar sequestration in soil

Table S28. The carbon price required to achieve the global temperature control target under the framework of the Paris Agreement ^{42,85}

Source	Price (USD ₂₀₁₀ /t)	Year	Target
IPCC, 2014	18-250	2020	2°C
IPCC, 2018	15-6050	2030	1.5°C

Table S29. Energy consumption of different transport models^{86,87}

Number	Power sources	Consumption intensity		Unit	Density (kg/L)	Price		Unit
		Current	From 2030			Current	From 2030	
1	Diesel	0.050	-	L/km	0.83	1.10	-	USD/kg
2	Gasoline	0.059	-	L/km	0.75	1.47	-	USD/kg
3	Hydrogen	-	0.007	kg/km	-	-	4.84 ^a	USD/kg
4	Electricity	-	0.121	kWh/km	-	-	0.06 ^b	USD/kWh

^{a.} Average price from reference⁸⁸

^{b.} Is equal to the domestic electricity price

Table S30. Net life-cycle GHG emissions of different transport distance and models

Transportation	GHG emissions (t CO ₂ -eq)	Net life-cycle GHG emissions (g CO ₂ -eq/MJ)	Proportion in whole system (%)	
20 km	Diesel	2.86×10 ²	-53.46	4.12
	Gasoline	4.06×10 ²	-52.59	5.74
	Hydrogen	2.12×10 ²	-54.00	3.08
	Electricity	8.51×10 ¹	-54.92	1.26
40 km	Diesel	5.73×10 ²	-51.38	7.92
	Gasoline	8.12×10 ²	-49.65	10.86
	Hydrogen	4.24×10 ²	-52.46	5.98
	Electricity	1.70×10 ²	-54.30	2.49

100 km	Diesel	1.43×10^3	-45.15	17.69
	Gasoline	2.03×10^3	-40.82	23.34
	Hydrogen	1.06×10^3	-47.86	13.71
	Electricity	4.26×10^2	-52.45	6.00

6.3 Sensitivity analysis of outcomes from scenarios for China's bio-NETs deployment

Based on the results as mentioned above, for a single BIPP plant, the impact of N₂O reductions on the total life-cycle GHG emission is small (between -1.0% and 3.0%), while the variations in biochar stabilities and transport distance/models play an important role in the BIPP life cycle GHG emissions (with change rates of -19.7% to 34.6%, and -6.9% to 10.6%, respectively). Thus, the range of cumulative GHG reductions from a change in biochar stability and transport distance/models is studied further in several potential scenarios for future bio-NETs deployment in China.

The results in Fig. S11 show that the cumulative GHG reduction by 2050 could be affected to a certain extent (-3.9% to 5.9% relative to the base case) under the scenarios 1-4 and 6-7. Furthermore, it shows that under scenario 5 (assuming deployment of BIPP only), the cumulative GHG reduction by 2050 is more sensitive (-11.3% to 17.0% relative to base case), compared with other scenarios. In this sense, BIPP coordinated with BECCS deployment will result in a lower uncertainty in future cumulative GHG reduction by 2050, compared to deployment of BIPP alone (scenario 5).

For a mid-term perspective, the initial highest biomass input in BIPP coordinated with further BECCS deployment (scenario 4) can reach up to 8337-9127 Mt CO₂-eq reduction by the end of 2050. To put this in a worldwide context, the global emissions reduction target required to meet the Paris Agreement goal of limiting the increase in global temperatures to no more than 1.5 °C by 2050 using BECCS technology was pegged at 28-65 Gt CO₂-eq⁵⁹. China by itself can achieve around 14-33% of the global removal goal by focusing strongly between now and 2050 on deployment of bio-NETs systems employing only agricultural, forest residues and energy plants (explained in SI 5.5). Furthermore, the scenario using 100% of agricultural residues, energy crops and forest residues (scenario 7) could provide cumulative reduction potentials of 21018 to 22674 Mt CO₂-eq by 2050. To sum up, these options under the “moderate” or “maximum” scenarios (aggressive deployment) could contribute to 175-3599 Mt CO₂-eq of GHG reductions by 2030. These value can be translated to a reduction on carbon emissions per unit of 2005 GDP by 2%-69%, which could approach or even achieve the goals alone (i.e., 60%-65% CO₂ reduction) included in China's NDC for the Paris Agreement⁶³.

Fig. S10. The sensitivity analysis of economic feasibility (left) and life-cycle GHG emissions (right) for a demonstration BIPP system

Fig. S11. The sensitivity analysis of outcomes from scenarios for China's bio-NETs deployment

(2) The key parameters of BECCS scenarios were not made clear in the document. Nowhere did the authors state the type of technology which would be utilised in this case (biomass combustion followed by CO₂ capture in amines with compression and underground CO₂ storage?). It was also unclear whether the ‘transition’ to BECCS meant a full technology transition from BIPP to BECCS, meaning full infrastructure transition for technology, or if BIPP would contribute but with flue gas outlets retro-fitted with CCS technology? In fact this would be an interesting additional scenario to investigate or perhaps could be explored in the maximum potential emission reduction scenario.

Response:

We appreciate the reviewer’s comments very much. In the revision, we have clearly described the key parameters of BECCS scenarios in both manuscript (section: Scenario analyses of BIPP technology deployment) and Methods.

The type of BECCS technology which would be utilized in this case is biomass gasification power system followed by CO₂ capture in mono ethanol amines with compression and underground CO₂ storage.

For the question “Whether the ‘transition’ to BECCS meant a full technology transition from BIPP to BECCS, meaning full infrastructure transition for technology, or if BIPP would contribute but with flue gas outlets retro-fitted with CCS technology”, the answer is: In scenario 2, 3, 4, 6 and 7, BIPP systems will be gradually transformed into BECCS systems directly just through reconstructions of part of equipment (e.g., reactors), changes of key reaction parameters (e.g., temperature, reaction atmosphere) and flue gas outlets retro-fitted with CCS technology. And for the BIPP systems that have not been transformed into BECCS in these scenarios, we have already considered that the CCS technology will be used in these BIPP system after 2030. It is assumed that the CO₂ produced from power generation and biomass fuel combustion is captured in mono ethanol amines, compressed and stored underground (BIPP+CCS) from 2030. However, to explore the situation without CCS development (in case of CCS failure), in scenario 5 we only focus on BIPP from 2020 to 2050. That is to say, BIPP+CCS has not been considered in scenario 5.

In the maximum potential emission reduction scenarios (scenario 6 and 7), our purpose is to find the maximum value in the extreme scenarios. Considering that BIPP+CCS will not capture the CO₂ emitted by the subsequent use of liquid products (bio-oil combustion as liquid fuel for vehicles), BECCS theoretically would capture more CO₂ than BIPP+CCS. Thus, it is designed in scenario 6 and 7 that BIPP will be transformed to BECCS after 2030 gradually, and the rest BIPP will be coupled with CCS. Also, there is no need to set another extreme scenario of only BIPP (2020-2030) and BIPP+CCS (2030-2050), as its cumulative GHG emission reduction must be lower than that of scenario 6 and 7.

Corresponding change in text (changes are highlighted in yellow):

In main text (manuscript: Scenario analyses of BIPP technology deployment):

The seven scenarios of BIPP deployment are developed and can be divided into two groups: the first “Moderate development of bio-NETs” (scenarios 1-5) includes options assuming bio-NETs deployment in different sustainable biomass allocations considering China’s use practices and policies for the collection and utilization of crop residues; the second “Maximum bio-NETs potentials” (scenarios 6-7) considers more ambitious options assuming optimal conditions aimed at maximizing China’s contributions to climate change mitigation by exploiting the full potential of its sustainable biomass resources in biochar and biofuels (Table 2, explained in SI 5 in detail). Note that scenarios 2-4 and 6-7 assume coordinated development of BIPP and BECCS after 2030: under these scenarios, a share of BIPP systems will be transformed into BECCS systems directly through retrofitting, changes of key reaction parameters, and CCS technology added to flue gas outlets; the remaining BIPP systems not yet transformed into BECCS systems will be coupled with CCS after 2030 (as explained in detail in Methods). The BIPP systems in these scenarios are based on the system analyzed above, including residue cultivation, transportation and biochar application.

In main text (Method part):

Scenarios. Based on the current state of China’s utilization of crop residues and prospects for exploitation of bio-energy NETs (BIPP and BECCS), two groups of scenarios (“Moderate” and “Maximum”) based on sustainable biomass availability in China were assumed. As defined

scenario-by-scenario in Table 2, the first group (moderate scenarios 1-5) assumes realistic levels of biomass availability between 2020 and 2050 and is labeled “Moderate development of bio-NETs”; the second group (maximum scenarios 6-7) assumes very high levels of biomass availability and is labeled “Maximum bio-NETs potentials”.

In scenario 1, 2, 3, 4, 6 and 7, based on current state-of-the-art research^{12,13} and in accordance with scenarios analyzed in IPCC reports summarized by Kemper⁷, it is assumed that BECCS technologies can be deployed on a large scale starting gradually in 2030. On the other hand, biochar production using BIPP technology is arguably readily available for deployment now and likely to be promoted in the near future in China. For these reasons, the scenarios cover two sequential time periods, including 2020-2030 when only BIPP will be available for deployment and 2030-2050 when both BIPP and BECCS will be available. In addition, to explore the situation without CCS development (in case of CCS failure), scenario 5 only focuses on BIPP (without CCS) from 2020 to 2050. Analysis of these scenarios enables us to identify preferable courses of action for the near-term deployment of biochar technologies to maximize GHG emissions reduction, providing critical and timely inputs for policy makers designing strategies and policies affecting bio-energy NETs.

Sustainability criteria. In consideration of multiple use paths for crop residues in China, the criteria for sustainable biochar production in the baseline scenario (scenario 3 in Table 2) requires that the biomass feedstock used in pyrolysis accounts for 33% of collected crop residues over the period 2020-2030, a percentage corresponding to the residues currently used as fuel (mainly for household cooking and heating) or burned directly in the field. Thus, the national pyrolysis application of the baseline scenario does not require a change in the amount of total biomass utilization, with residues producing energy supply (i.e., the electricity and pyrolysis gas) from the BIPP systems. It is further assumed in additional moderate scenarios (numbers 1, 2, 4) that the BIPP feedstock for 2020-2030 is 0%, 11% and 73% of the total current availability of crop residues, respectively, and by 2050 it could achieve 80% by retaining the minimum of 20% required for livestock feed (rationales for these percentages are explained in SI 5.1).

For scenarios 2-4 and 6-7 in the period 2030-2050, this study assumed that BECCS is by then mature and can be developed gradually in coordination with BIPP. Biomass resource will be used both in newly built BECCS plants and retrofitted plants from BIPP system.

First, it is assumed that BECCS is deployed from 2030 onward and the biomass used in BECCS increases steadily through gradual deployment, rising to 69% (in scenario 2), 47% (in scenario 3), and 7% (in scenario 4) by 2050, respectively. In addition, because of the superior carbon sequestration potential of BECCS, it is assumed that the biomass used in BIPP systems will then be gradually transitioned through retrofitting of plants and changes in key reaction parameters, at annual rates of 0.74%, 2.7% and 13.0%, respectively. The calculation equation is as follows:

$$R_i = \sqrt[20]{\frac{Per_{target}}{Per_i}} - 1 \quad (6)$$

where R_i indicates the annual rate of increase of biomass used in BECCS for scenario i^{th} (scenario 2 to 4), in which the biomass increases from the BIPP system transformation; Per_{target} and Per_i are the percentages of crop residues in BECCS in 2050 and 2030 respectively, and, in this study, it is assumed that the targeted percentage would be 80% of residues in BECCS. The percentage for each scenario is explained in SI 5.1 and 5.2.

For scenarios 6-7 exploring the maximum bio-NETs potentials, the BIPP systems would be developed by making use of all biomass (100% of crop, forest residues, and energy crops) during 2020-2030. All of the BIPP plants would then be transitioned to BECCS gradually with an annual average change rate of 5% in the period 2030-2050. In these scenarios, the BIPP systems that have not been transformed into BECCS will be coupled with CCS technology after 2030. It is assumed that the CO₂ produced from power generation and biomass fuel combustion is captured by mono ethanol amines, compressed and stored underground (BIPP+CCS) from 2030 onward.

(3) The BIPP process is not well described in the supplementary information regarding the ASPEN model used and results were somewhat difficult to interpret. Was the BIPP process

optimised to ensure maximum energy efficiency/recovery? In particular the internal use of energy (including heat exchanger parameters) were difficult to interpret from information provided. The authors did not attempt to compare the transformation or electric efficiency to any other published works regarding slow pyrolysis, making it difficult to gauge if their final energy efficiency values are reasonable. The ASPEN model diagram also appears to be incorrect and does not match with the discussion. For example, after exiting the pyrolysis reactor the vapour stream is water cooled, with the hot water outlet mixed with an air stream (HEATRECO) and then referred to as 'dry-air', the heat of which is claimed for drying. If this is an indirect contact heat exchanger this would be a very inefficient way to dry incoming biomass, but direct drying wouldn't be possible either since introducing moisture in the dryer would not be desirable. The heat flows are also shown seemingly in the wrong direction in Figure S1, pointing towards both HEAT-WST and HEATSUPP instead of away. In Section 1.2.3 (sup) SEC-EXCH is said to further cool pyrolysis vapours using 'cold water', however this water is directed from VAT-COOL (hard to read label in figure provided) and is labelled as hot water being the inlet to SEC-EXCH, cold water being the outlet which would suggest the vapour mix temperature would increase, making it a heater not cooler. AIR-PRH is also unclear showing two air inlets and outlets of unknown temperatures and no way of knowing which way heat was transferred in this unit operation. A property table for an example process basis (1tph biochar production for example) including streams flows and properties (including temperatures) would make the ASPEN model more accessible and able to be interpreted. As it stands it is difficult to judge whether the BIPP process is well designed or optimized. For example, why did the authors decide to burn biomass to make heat? Would it not be more efficient to use internally generated syngas in a burner to achieve a higher heat transfer efficiency? The use of cooling water is also not well justified. Air cooling would negate the need for water treatment, discussed by the authors to have significant greenhouse gas implications for the life cycle analysis. Of course, air would have a reduced heat transfer rate compared to water, however the authors did not discuss or describe whether this benefit is balanced by the need to process water for use in water cooling. Did the authors consider recycling of water within the plant? On p13 line 249 30% of GHG emissions attributable to the plant are from waste water treatment – but this should be a very small amount of make-up water

presumably if it is able to be recovered internally. The inlet moisture of the feedstock is also a major determinant for the overall energy balance of a slow pyrolysis process since it demands high energy for drying, how was the feedstock moisture determined/decided? This needs better discussion/acknowledgement in the paper and may be an important overarching assumption to do sensitivity analysis around in terms of process modelling outcomes.

Response:

We would like to thank the reviewer very much for these detailed and valuable comments. In response to this comment, we have modified the Aspen model and clarified the block parameters and stream properties used in the model. The simulation results have been changed a little in the revision (mainly due to the increased auxiliary power use from bag filter). We thus updated the results. And a property table (Table S1) has been added in Supplemental Information.

1. Firstly, we want to response the doubts about BIPP optimization and comparison with other published works. In short, the BIPP process has been optimized to ensure higher energy efficiency/recovery, while it was validated by data of onsite observation and inspection. Before the BIPP technology deployment, we have done lots of experiments and analysis to explore the pyrolysis mechanism and achieve process optimization, and have authored many papers on related study that have been published in energy journals [1-7]. We thus designed its Aspen model based on the achievements from experimental study in laboratory, and apply it in a demonstration plant in Ezhou, Hubei province. Then based on the experiments results and actual data of BIPP system, the Aspen simulation has been validated (SI 1.4) to ensure the system reliability including reaction products and system energy efficiency etc. It is worth mentioning that this BIPP technology has won the Blue-Sky Award from the United Union as one of 2014 global top investment scenarios to apply new technologies for renewable energy utilization. For comparison with other published works, we have added in Supplementary Information (Table S8). The comparison results show that the energy efficiency is similar to that observed by other researchers when all processes of the pyrolysis system are fully considered. Because this technology is very new

and mainly focus on the experiments and application level, there is lack of similar published works about BIPP system assessment.

2. Secondly, for Aspen model, we apologize for our carelessness and thank the reviewer for pointing this out. In revision, we have checked and modified the model in a reasonable way according to the data from experiments, as shown in the revised Fig. S1. A detailed table has been added, including streams flows and properties (Table S1), so as to interpret the Aspen BIPP model more clearly as reviewer's suggestion.
 - a. In the modified model, the "DRY-AIR" is composed of flue gas of biomass combustion and hot air (RECYAIR). And this is a direct drying, so that it could efficiently dry the biomass feedstock. The temperature of exhaust stream (EXHAUST) after heat exchange with biomass dryer is 110 °C, moisture in biomass will become water vapor and be taken away by exhaust stream, thus there is no introducing moisture in the dryer.
 - b. For heat flows, we accidentally marked the wrong direction for the arrows. In the revision, all the arrows have been checked and all the wrong directions have been revised.
 - c. The "COOLWAT" label has been changed to the "HOTERWAT", which will help to understand the BIPP processes in an easier way.
 - d. AIR-PRH is an air preheating unit. The AIR2 is heated to HOT-AIR2 (50 °C) by shell-and-tube heat exchange with HOT-AIR4. The inlet temperature of AIR2 and HOT-AIR4 is 25 °C and 104.2 °C, respectively.
3. Finally, we would like to thank the reviewer for the valuable comments. We also would like to explain that the studied BIPP system is well-designed so as to balance the economic performance and efficiency.
 - a. In the feasibility report of the BIPP demonstration plant, the heat needed in pyrolysis process is come from biomass combustion but not pyrolysis gas combustion. . It is because the lower heating value ($>10.3 \text{ MJ/Nm}^3 / 230.5 \text{ MJ/kmol}$) of pyrolysis gas is higher than the minimum CNS requirement ($>7 \text{ MJ/Nm}^3$) for fuel gas for urban residents (GB 50028-2006), which make it valuable. Based on our estimation, using pyrolysis gas to heat, will lower the system economy, especially in China. Compared

to the use of internally generated pyrolysis gas in a burner, the pyrolysis gas selling and biomass burning would bring more economic value.

- b. As for the cooling water (WATER), we choose water cooling rather than air cooling, which is according to the cooling requirement of system. The separated hot-temperature biochar is first cooled down by air and then by water, so as to guarantee the quantity of biochar. Then, the WATER is used in second-stage separator (water cooling tower), which is a shell-and-tube indirect heat exchange. A condensate collection channel is arranged at the bottom of the heat exchange. Thus, the light materials and part of condensed water from pyrolysis vapor could enter the light oil tank. Furthermore, the WATER output will be used in gas purification and absorption system, in which the water is sprayed to wash the pyrolysis gas and remove the trace tar contained. As the output of the gas purification and absorption system, the final WATER (as wastewater) would be treated in wastewater treatment system, and then emitted to the environment.
- c. We have considered that the final WATER output (as wastewater) would be treated in wastewater treatment system. However, limited by the Aspen block and in order to simplify the system, we have simulated the gas purification and absorption system (CYC-MIX, CYC-SPI, CYC-PUM and SPR-SEP) without washing water flow and wastewater treatment system. The gas purification and absorption system could deeply remove most dust, water as well as residual acid, alcohol and tar in pyrolysis gas. And we have calculated the quantity of wastewater in the life-cycle assessment and its associated GHG emissions. To make it clear, we have modified the Supplemental Information and have clearly interpreted the cooling water and wastewater treatment (SI 3.2.1.4 and Table S14) in the revision.
- d. Thank reviewer's valuable comment, the inlet moisture of feedstock is indeed an important parameter in traditional studies. **According to the feasibility report of the BIPP demonstration plant, the inlet biomass moisture for pyrolysis reactor is required to be controlled within 10.00 wt.%. In China, an industrial chain for biomass collection has already formed. The collectors collect the crop residues from scattered farmers in a wide area, then air dry the biomass and process them (chopped**

and baled), so as to provide biomass to plants in a flexible and convenient way. The biomass moisture after air drying ranges from 2.36-16.32 wt.% (rice 5.93 wt.%, maize 8.4 wt.%, wheat 7.3 wt.%, bean 2.36 wt.%, cotton 5.02 wt.%, peanut 16.32 wt.%, rape 9.64 wt.% and sesame 7.91 wt.%) [1]. For the worst condition (peanut 16.32 wt.%), 1.58×10^5 kJ energy is needed for reducing the moisture of 1 t peanut residues from 16.32 wt.% to 10.00 wt.%. According to the results of Aspen model, the mass flow of DRY-AIR is 8120 kg/h, the temperature of DRY-AIR and EXHAUST stream is 171 °C and 110.4 °C, respectively. Thus, the 60 °C temperature difference could provide 2.94×10^5 - 3.92×10^5 kJ energy when the efficiency of heat exchange is about 0.6-0.8. These energy from temperature difference could totally provide the needed energy for drying highest moisture biomass (1.58×10^5 kJ energy requirement). To sum up, although the inlet moisture of the feedstock is an important parameter in pyrolysis process, the moisture could be controlled in an acceptable range by heat exchange of exhausted flue gas (DRY-AIR). Thus, we do not conduct a sensitivity analysis on feedstock moisture.

Reference:

- [1] Yang H, Yan R, Chen H, et al. Characteristics of hemicellulose, cellulose and lignin pyrolysis [J]. Fuel, 2007, 86(12-13): 1781-1788.
- [2] Chen Y, Yang H, Wang X, et al. Biomass-based pyrolytic polygeneration system on cotton stalk pyrolysis: influence of temperature [J]. Bioresource Technology, 2012, 107: 411-418.
- [3] Xin S, Yang H, Chen Y, et al. Assessment of pyrolysis polygeneration of biomass based on major components: product characterization and elucidation of degradation pathways [J]. Fuel, 2013, 113: 266-273.
- [4] Yang H, Yan R, Chen H, et al. Pyrolysis of palm oil wastes for enhanced production of hydrogen rich gases [J]. Fuel Processing Technology, 2006, 87(10): 935-942.
- [5] Chen Y, Zhang X, Chen W, et al. The structure evolution of biochar from biomass pyrolysis and its correlation with gas pollutant adsorption performance [J]. Bioresource Technology, 2017, 246: 101-109.

[6] Gao Y, Wang X, Wang J, et al. Effect of residence time on chemical and structural properties of hydrochar obtained by hydrothermal carbonization of water hyacinth [J]. Energy, 2013, 58: 376-383.

[7] Chen H, Lin G, Chen Y, et al. Biomass pyrolytic polygeneration of tobacco waste: product characteristics and nitrogen transformation[J]. Energy & Fuels, 2016, 30(3): 1579-1588.

Corresponding change in text:

In Tables section (in Supplementary Information):

Table S1. Aspen Plus unit operation block description

Block name	Block parameters	Description
(Aspen block)		
DRYER (Ryield)	Pressure=1 atm; T=150 °C	Separates the water in conventional components
PYR-DECO ((Ryield)	Pressure=1 atm; T=250 °C	Separates the subcomponents (including cellulose, hemicellulose, lignin and extractives)
FIR-EXCH (HeatX)	Hot stream outlet temperature=150 °C	Cooling down the hot vapor from pyrolysis reactor by the cold flue
FIR-SEP (Flash2)	Pressure=3 bar; T=150 °C	Separate the heavy condensable (such as Levoglucosan and Free fatty acids) in hot vapor
SEC-EXCH (HeatX)	Hot stream outlet temperature=50 °C	Cooling down the vapor from first separator by the cold water
SEC-SEP (Flash2)	Pressure=3 bar; T=50 °C	Separate the light material (such as Glycol-aldehyde and CH ₃ OH)
CYC-MIX (Mixer)	Pressure=1 bar	Mix the crude pyrolysis gas with separated stream (mainly recycling water) from SPR-SEP
CYC-PUMP (Pump)	Discharge pressure=0.5 bar	Pump separated stream from SPR-SEP
SPR-SEP (Flash2)	Pressure=3 bar; T=25 °C	Separate the tar and acid of the pyrolysis gas
Boiler (RGibbs)	Pressure=0.8 atm	Pyrolysis gas combustion process
Turb (Compr)	Discharge pressure=0.15 bar	Turbine steam expansion
AIR-PRH2 (HeatX)	Hot stream outlet temperature=100 °C	Air preheating by fume from gas turbine
COOL-AIR (Compr)	Discharge pressure=0.8 atm	Air compression
AIR-COOL (HeatX)	Hot stream outlet temperature=150 °C	Biochar cooling by cooled air
WAT-COOL (HeatX)	Hot stream outlet temperature=100 °C	Biochar cooling by cooled water
AIR-BURN (Compr)	Discharge pressure=0.8 atm	Air compression
AIR-PRH (HeatX)	Hot stream outlet temperature=50 °C	Air preheating by hot fume heated by biochar
FUEL-DEC (RStoic)	Pressure=1 atm; T=25 °C	Separate the constituent elements based on the biomass ultimate analysis

FUEL-COM (RGibbs)	Pressure=1 atm	Biomass fuel combustion process
HEATSUPP (Heater)	Pressure=1 atm	Heat exchange providing heat to pyrolysis reactor
HEAT-WST (Heater)	Pressure=1 atm	Heat exchange providing heat to biomass dryer
Materials and stream flows for 1 t rapeseed biomass feedstock input at 650 °C		
S-FUEL	Value=0.72 t/h; Pressure=1 atm; T=25 °C	Biomass used as combustion fuel
AIR1	Value=210 kmol/h; Pressure=1 atm; T=25 °C	Air used in biomass combustion
AIR3	Value=50 kmol/h; Pressure=1 atm; T=25 °C	Air used to cool biochar down
HOT-AIR4	Value=50 kmol/h; Pressure=0.81 atm; T=104.2 °C	Hot air after heat exchange with biochar
AIR5	Value=160 kmol/h; Pressure=1 atm; T=25 °C	Air used in pyrolysis gas combustion
EXHAUST2	Value=166.2 kmol/h; Pressure=0.15 bar; T=100 °C	Exhaust stream after heat exchange with air
WATER1	Value=600 kg/h; Pressure=1 atm; T=25 °C	Water used to cool down
HOTERWAT	Value=600 kg/h; Pressure=1 atm; T=81.3 °C	Water output after heat exchange with pyrolysis vapor
C-FUME	Value=7662.1 m ³ /h; Pressure=1.0 bar; T=135.4 °C	Exhaust fume after heat exchange with pyrolysis reactor
RECYFUME	Value=8580.8 m ³ /h; Pressure=1.01 bar; T=184.4 °C	Heated vapor water after heat exchange with pyrolysis vapor
EXHAUST	Value=8765.7 m ³ /h; Pressure=1.0 bar; T=110.4 °C	Exhaust stream after through the bag filter (B1)

Table S8. Energy performance of different technologies^{12,65,66}

Study	Process type	Description	Energy efficiency
This study	Intermediate pyrolysis	A BIPP demonstration plant (250-650 °C)	48.75% - 73.69%
1	Slow pyrolysis	A continuous pyrolysis poly-generation system (550-650 °C)	82.1%
2	Slow pyrolysis	A pilot pyrolysis plant (550 °C)	58.9%
3	Fast pyrolysis	The pyrolysis system, including hot vapor filtration, a fractional condenser, and an electrostatic precipitator (500 °C)	61.01%

In Figures section (in Supplementary Information):

Fig. S1. Flowsheet for the biomass intermediate pyrolysis poly-generation process

In Supplemental Information

3.2.1.4 Greenhouse gas emissions from the demonstration BIPP plant

In this study, the wastewater is mainly comprised of cooling water and domestic water. Cooling water is first used to decrease the temperature of the biochar, is then used in the second-stage separator (water cooling tower), and finally goes into the gas purification and absorption system. The second-stage separator is a shell-and-tube indirect heat exchanger. A condensate collection channel is arranged at the bottom of the heat exchanger, so light materials and part of the condensate water of pyrolysis gas can enter the light oil tank. In the gas absorption system, water is used to wash the pyrolysis gas so as to remove the contained trace tar. The final water output is wastewater and processed in a wastewater treatment system. Limited by the Aspen block and in order to simplify the system, this study has simulated the gas purification and absorption system (CYC-MIX, CYC-SPI, CYC-PUM and SPR-SEP) without washing water flow and the wastewater treatment system. Domestic water is daily water consumption by workers, which is ignored in the system diagram.

Nevertheless, the volume of the treated wastewater (cooling water and domestic water) is

calculated based on the feasibility report. Based on available data and equations, the CH₄ emissions associated with wastewater treatment are 37.0 t/yr (924 t CO₂-eq/yr).

Table S14. The CH₄ emissions associated with wastewater treatment for a BIPP system with 7.5 t biomass feedstock input per hour

Wastewater items	Quantity (kg/h)	COD (kg/L) ^a	CH ₄ emissions (t/yr)
Cooling water	4650	0.0125	36.5
Domestic water	2460 ^a	0.0003	0.5
Total	7110	-	37.0

^a From feasibility report¹¹

Other specific comments include:

MAIN PAPER

Line 26: Deployment of low or zero emission technologies, rather than ‘deployment of emission technologies’?

Response:

We would like to thank the reviewer’s comment. We have modified the sentence in Abstract.

Corresponding change in text (in Abstract):

Deployment of negative emission technologies needs to start immediately if we are to avoid overshooting international carbon targets, reduce negative climate impacts, and minimize costs of emission mitigation.

Line 183 and 184: Energy efficiency is said to decline gradually with the increase in pyrolysis temperature, however Fig. 1 shows ‘electric power efficiency’ which actually increases with temperature (along with increase in syngas generation, it is guessed). Presumably the authors are talking about the energy efficiency defined in Table S6 in these lines, however this is not clear. In fact, it is not clear why the electric efficiency is displayed in Fig 1 instead of energy efficiency (since it is not discussed in the text) nor how electric efficiency value is calculated

in the work. Electrical energy output divided by total input feedstock LHV (including biomass combusted??)?

Response:

We appreciate the reviewer's comments. Both energy efficiency and electric power efficiency are important indicators. We apologize that we did not describe them clearly. According to reviewer's comments, we have added the explanations in main text for system gross/net electric efficiency in Fig. 1 (Manuscript: Dynamic simulation of a BIPP system), and have added the explanation for system energy efficiency in Supplementary Information (SI: 1.6 Analysis of BIPP energy flows).

The system energy efficiency is defined as the useful energy that is recovered from pyrolysis products (i.e., pyrolysis gas, electricity, bio-oil and biochar) divided by the total energy input (including all combusted and pyrolyzed biomass and consumed electricity). The system gross/net electric efficiency is defined as the gross/net power generation divided by the energy input from pyrolysis gas.

The energy efficiency declines gradually with the increase in pyrolysis temperature as has been shown in Table S7, which is mainly connected to the larger energy consumed and change of production prosperities. The gross/net energy efficiency has been shown in Fig. 1: the gross electrical energy efficiency is around 30% with small change; the net energy efficiency has a large increase during 250°C to 350°C, and then is stable around 25%, which is mainly connected to the changes of auxiliary power that used for the whole BIPP systems.

Corresponding change in text:

In the main text (Dynamic simulation of a BIPP system)

Fig. 1 Performance of a BIPP system with temperature varying from 250°C to 650 °C for 1 t biomass input per hour: a) Material flows: fuels input for high-temperature flue gas generation, cooling water inputs used to cool biochar, and the biochar output; b) Electric power flows: electricity generation by the pyrolysis gas combustion power system, electricity consumption of the whole system, and net power generation; c) Electric power efficiencies: gross/net efficiencies of electricity generation in the BIPP system, which are defined as the gross/net power generation divided by the energy input from pyrolysis gas.

In the Supplementary Information (1.6 Analysis of BIPP energy flows)

The energy inputs to the BIPP system are calculated based on the biomass feedstock input for pyrolysis and heat provision, as well as the electrical energy supplied to the system. Energy outputs are the sum of the energy content of the pyrolysis products (biochar, pyrolysis gas and bio-oil) and the energy loss during plant operation. The system energy efficiency is calculated

for the energy transformation, which is defined as the useful energy recovered from pyrolysis products (i.e., pyrolysis gas, electricity, bio-oil and biochar) divided by the total energy input.

Figure 1: It would be useful on Fig 1 a) to also show biochar generation rate to explain the change in cooling water usage as a trade-off between reaction temperature and amount of biochar produced. It is also unclear why the authors considered such low temperatures as 250°C here where the biomass would not be pyrolyzed at all but rather torrefied, and certainly not stabilized at this stage. It is strange the authors chose this parameter (temperature) to investigate when there are many other process variables which would be more likely to change (including feedstock inlet moisture for example).

Response:

Thanks for reviewer's suggestion. We have modified Fig. 1 and added the biochar generation rate (kg/h) in Fig. 1 a).

- a. For question on pyrolysis temperature of 250°C, we agree that most components of biomass will not pyrolysis at such a low temperature. However, certain special biomass with large content of hemicellulose may pyrolyse even at such low temperature, as it is commonly perceived that hemicellulose begins to decompose between 210 °C to 320 °C [5]. In many studies, the pyrolysis temperature for common agricultural stalks and woods ranges from 250 °C to 650 °C (e.g., Reference [2-4]). Thus, in our study, we have considered such low temperature as 250 °C to investigate the energy and economic performance of BIPP system.
- b. For question on choosing pyrolysis temperature as key parameter, we would like to explain that pyrolysis temperature is indeed the key parameter of reaction conditions in the whole thermochemical conversion of biomass. In real operation, temperature is also the key parameter that the operator can control to adjust the pyrolysis process. And both the distribution and properties of pyrolysis products and system energy efficiency will be influenced largely by pyrolysis temperature [6]. Other process variables such as feedstock inlet moisture should be controlled within a certain range (<10% in this study) by the drying or torrefaction technology. Thus, temperature is chosen as the key parameter in this study.

Reference:

- [1] Van der Stelt M J C, Gerhauser H, Kiel J H A, et al. Biomass upgrading by torrefaction for the production of biofuels: A review [J]. Biomass and Bioenergy, 2011, 35(9): 3748-3762.
- [2] Agrawal R K. Kinetics of reactions involved in pyrolysis of cellulose II. The modified kilzer - bioid model [J]. The Canadian Journal of Chemical Engineering, 1988, 66(3): 413-418.
- [3] Broido A, Nelson M A. Char yield on pyrolysis of cellulose [J]. Combustion and Flame. 24: 263-268, 1975, 24: 263-268.
- [4] Scheirs J, Camino G, Tumiatti W. Overview of water evolution during the thermal degradation of cellulose [J]. European Polymer Journal, 2001, 37(5): 933-942.
- [5] Gao Y, Wang X, Chen Y, et al. Pyrolysis of rapeseed stalk: Influence of temperature on product characteristics and economic costs [J]. Energy, 2017, 122: 482-491.
- [6] Kan T, Strezov V, Evans T J. Lignocellulosic biomass pyrolysis: A review of product properties and effects of pyrolysis parameters [J]. Renewable and Sustainable Energy Reviews, 2016, 57: 1126-1140.

Corresponding change in text (in manuscript):

Fig. 1 Performance of a BIPP system with temperature varying from 250°C to 650 °C for 1 t biomass input per hour: a) Material flows: fuels input for high-temperature flue gas generation, cooling water inputs used to cool biochar, and the biochar output; **b)** Electric power flows: electricity generation by the pyrolysis gas combustion power system, electricity consumption of the whole system, and net power generation; **c)** Electric power efficiencies: gross/net efficiencies of electricity generation in the BIPP system, which are defined as the gross/net power generation divided by the energy input from pyrolysis gas.

Line 272-274: Authors suggest it is possible to do a like-for-like replacement of fossil fuels with pyrolysis products, but do not seem to directly compare the heating value of the produced products with expected replacements. It is very unlikely that the syngas would be usable as a direct natural gas replacement, and would likely have a much lower heating value. Did the authors account for this? The method is not clearly described in this case.

Response:

We would like to thank the reviewer for their helpful comments. We totally agree that heating value of products should be considered. According to reviewer's comment, in the revision pyrolysis products (bio-oil and pyrolysis gas) are transferred into fossil fuels (coal tar and coke oven gas) based on energy equivalent. To make the description more clearly and estimation more reasonable, we also have made the following changes in the revision.

- a. **To describe the gas product more accurately, we have modified the whole manuscript changing the “biogas” to “pyrolysis gas”.** Pyrolysis gas contains a significant amount of carbon monoxide, hydrogen, along with methane and carbon dioxide, may be used as a fuel for industrial combustion. Syngas is the short name for a gasification product known as synthesis gas, which mixes of hydrogen, carbon monoxide, and carbon dioxide and could be used as a potential intermediate in the conversion of certain biomass in fuel. Although both of them have similar basic components, pyrolysis gas has much higher heating value due to the high contents of combustible gases (e.g., CH₄) [1].
- b. For expected replacement of pyrolysis gas, we agree with the reviewer that the lower heating value (LHV) of natural gas (about 33-41 MJ/Nm³ in China) is usually higher than that of pyrolysis gas (about 14MJ/Nm³ in this study). Also, in the current situation of natural

gas scarcity in China, households in rural areas usually use coal gas (e.g., coke oven gas, LHV is around 9 MJ/Nm³ [3]) instead of natural gas. Considering that the BIPP system would usually be built in rural areas, we have considered the similarities (e.g., LHV and main components) of pyrolysis gas and coke oven gas, and modified the expected replacement of pyrolysis gas from natural gas to coke oven gas, which is also mainly composed of CH₄ and H₂ and CO. The heating value of the pyrolysis gas from the BIPP system is always higher than the minimum requirement of Chinese National Standard (CNS) (>7 MJ/Nm³) for fuel gas for urban residents (GB 50028-2006) [4]. According to the comments from reviewer, we have modified the Supplemental Information and described them in a more clearly way.

Reference:

- [1] Trabelsi A B H, Ghrib A, Zaafouri K, et al. Hydrogen-rich syngas production from gasification and pyrolysis of solar dried sewage sludge: experimental and modeling investigations[J]. BioMed Research International, 2017, 2017.
- [2] Nie H, Yang Y and Zhang P. Change of the Spatial Pattern of Rural Energy Consumption in China [J]. China Population, Resources and Environment, 2010, 20(04):29-34.
- [3] Liu X, Yuan Z. Life cycle environmental performance of by-product coke production in China [J]. Journal of Cleaner Production, 2016, 112: 1292-1301.
- [4] Gao Y, Wang X, Chen Y, et al. Pyrolysis of rapeseed stalk: Influence of temperature on product characteristics and economic costs [J]. Energy, 2017, 122: 482-491.

Corresponding change in text (in Supplementary Information):

4.2 Avoided air pollution from BIPP system

The avoided air pollution from BIPP system is the avoided air pollution from replacement of fossil fuels with BIPP products, subtracting the air pollution emissions from the combusted biomass stream used to generate heat for pyrolysis in BIPP system.

The BIPP system produces pyrolysis gas, bio-oil and biochar, the last sequestered into soil. For the pyrolysis gas, the LHV (shown in Table S6) is higher than the minimum one required by CNS requirement (>7 MJ/Nm³) for fuel gas for urban residents (GB 50028-2006). Considering

the LHV and main components (i.e., CO, H₂ and CH₄), pyrolysis gas is similar with those of coke oven gas (COG, about 9 MJ/Nm³), it is assumed that the pyrolysis gas could replace the COG for use by China's residents especially in the countryside. For bio-oil, it can be used as an alternative to coal tar after further purification and separation.

Because pyrolysis technology allows for removal of most particulates, mercury, and nitrogen and sulfur compounds, the utilization of pyrolysis gas in power generation and household heating/cooking can effectively reduce air pollutant emissions compared to supercritical coal-fired power plants (Sub-PC), combustion of COG and crude coal tar.

The air pollution emissions associated with the products of BIPP are allocated using a market-value method. The market prices are collected from a previous feasibility report¹⁴, with the allocation shown in Table S20. As a result, the emission reduction for air pollutant k by BIPP (M_{BIPP}^k) can be calculated as follows:

$$M_{BIPP}^k = (C_i^k - C_{BIPP}^k) \cdot E_{BIPP} - P_{bio-fuel} \times C_{bio-fuel}^k \quad (13)$$

where C_i^k , C_{BIPP}^k are the emission factors for air pollutant k associated with production of one unit of electricity, COG, coal tar, pyrolysis gas, and bio-oil (Table S21); i represents respectively supercritical coal power plants, COG, or coal tar sources; E_{BIPP} represents the products obtained by the BIPP (electricity, pyrolysis gas and bio-oil); and $C_{bio-fuel}^k$ indicates the emission factors for air pollutant k associated with combusted biomass fuel (unit: kg/t biomass fuel input) in BIPP system (data are from the feasibility report of the demonstration BIPP plant¹¹); $P_{bio-fuel}$ represents the biomass combusted to generate heat for pyrolysis, k indicates the following air pollution species: SO₂, NO_x, primary PM_{2.5} and BC.

Table S21. GHG and pollutant emission factors from power plants

	Supercritical coal-fired power plant (Sub-PC, g/kWh)	Coke Oven Gas (COG, g/Nm ³) ^a	Coal tar (g/t) ^a	Charcoal (g/MJ) ^b	BIPP systems			
					Electricity ^c (g/kWh)	Pyrolysis gas ^c (g/Nm ³)	Bio-oil ^c (g/t)	Emissions from the combusted biomass (kg/t) ^d
SO ₂	0.331	0.300	1098.070	-	0.289	0.357	309.660	0.320

NO _x	0.968 1.583)	(0.773-	0.106	388.700	-	0.819	1.012	877.240	1.020
PM _{2.5}	0.168 0.236)	(0.116-	0.178	651.360	-	0.073	0.091	78.850	9.730
GHG (CO ₂ - eq)	747.550		443.190	1622806.200	107.000	-	-	-	-
LHV	-		9.13 MJ/Nm ³	33.45 MJ/kg	-	-	13.69 MJ/Nm ³	5.93 MJ/ kg	

- a. Adopting energy content allocation method for coal tar which is a byproduct of coke production.
- b. The charcoal production is used in metallurgical and other industries, it can be substituted by biochar in alternative scenario (seen in Life-cycle GHG emissions for a demonstration BIPP system).
- c. The avoided air pollution from replacement of fossil fuels with BIPP products.
- d. According to the feasibility report¹¹.

Line 361: It is assumed that the BIPP process will drastically reduce particulates and other environmental pollutants, however the process outline did not show or consider any flue gas treatments, particularly for the combusted biomass stream used to generate heat for pyrolysis. Did the authors simply assume BIPP would result in no particulates, or was there a regulated standard applied to the BIPP process flue streams? NO_x may also be generated during pyrolysis when done under nitrogen as the inert atmosphere, this did not appear to be accounted for or discussed in the work.

Response:

We appreciate the reviewer’s comments very much. In the revised manuscript, a bag filter is added in Aspen Plus simulation to treat flue gas from the combusted biomass stream used to generate heat for pyrolysis. In the BIPP demonstration plant, the combustion gas after exchange heat with biomass in biomass drying sector is also transferred into bag filter and then emitted into atmosphere.

And the pyrolysis gas is also treated. In this BIPP system, after separated from two major vapor-liquid separators, the pyrolysis gas would be transferred into the purification tower, which is composed of mixer (CYC-MIX), circulating pump (CYC-PUM), flow divider (CYC-SPI) and gas separator (SPR-SEP). In this process, pyrolysis gas will be purified before its applications. In the actual BIPP demonstration plant, the remaining components (e.g., tar, acid) and part of

particulates (e.g., dust, heavy metals) from the pyrolysis gas will also be removed by an extra charcoal layer in purification tower.

And in later research about economic and environmental analysis, we have considered the whole BIPP plant including flue gas treatment according to the key data from demonstration plant.

For NO_x emission, we have estimated it for BIPP systems, including avoided NO_x from replacement of fossil fuels with BIPP product and the NO_x emissions from the combusted biomass stream used to generate heat for pyrolysis. We also compared it with NO_x emission from open biomass burning (OBB), and domestic biomass burning (DBB). The results have been shown in Table S18-20.

Corresponding change in text (in Supplementary Information):

Table S21. GHG and pollutant emission factors from power plants

	Supercritical coal-fired power plant (Sub-PC, g/kWh)	Coke Gas (COG, g/Nm ³) ^a	Oven (COG, g/Nm ³) ^a	Coal tar (g/t) ^a	Charcoal (g/MJ) ^b	BIPP systems			
						Electricity ^c (g/kWh)	Pyrolysis gas ^c (g/Nm ³)	Bio-oil ^c (g/t)	Emissions from the combusted biomass (kg/t) ^d
SO ₂	0.331	0.300	1098.070	-	0.289	0.357	309.660	0.320	
NO _x	0.968 (0.773-1.583)	0.106	388.700	-	0.819	1.012	877.240	1.020	
PM _{2.5}	0.168 (0.116-0.236)	0.178	651.360	-	0.073	0.091	78.850	9.730	
GHG (CO ₂ -eq)	747.550	443.190	1622806.200	107.000	-	-	-	-	
LHV	-	9.13 MJ/Nm ³	33.45 MJ/kg	-	-	13.69 MJ/Nm ³	5.94 MJ/kg	-	

^e. Adopting energy content allocation method for coal tar which is a byproduct of coke production.

^f. The charcoal production is used in metallurgical and other industries, it can be substituted by biochar in alternative scenario (seen in Life-cycle GHG emissions for a demonstration BIPP system).

^g. The avoided air pollution from replacement of fossil fuels with BIPP products.

^h. According to the feasibility report¹¹.

Figure 5: The grey bars are very difficult to read and interpret on the figure.

Response:

Thanks for the reviewer to point it out. We have revised the Figure 5 in an easier to read way.

Corresponding change in text (in manuscript: Air quality benefits due to nation-wide deployment of BIPP with biochar sequestration):

Fig. 5 Reductions in annual air pollutant emissions (SO₂, NO_x, PM_{2.5}, and BC) achieved by nation-wide deployment of BIPP systems with biochar sequestration. The color shading represents the levels of reduced emissions for different air pollutants. The grey labels indicate total emissions reductions by province resulting from three sources: BIPP substitution for other energy uses, avoided domestic biomass burning (DBB) and avoided open biomass burning (OBB). The BIPP scenario assumes use of 33% of available sustainable crop residues in China. The regional divisions are shown in Fig. S9, and the regional pollutant emissions abatement (%) is defined as the regional emissions reduction divided by the regional emissions in 2014.

Line 467/468: Here the authors claim that large capacity BIPP plants can result in additional positive environmental benefits but did not explain in comparison to what or via what mechanism? In comparison to small plants? Is there a limit on the size of BIPP plants

considered in the study? There will be a certain throughput for which a fully engineered process including auxiliaries would become completely impractical, this wasn't fully disclosed in the work.

Response:

Thanks for the reviewer to point out this error. In this study, the large capacity BIPP plants (larger biomass input) can result in additional economic (not environmental) benefits compared with small plant (smaller biomass input).

In this study, we have investigated the relationship between plant capacity (i.e., biomass input) and economic results by Aspen Process Economic Analyzer (APEA). However, we did not explore the limit on the size of BIPP plants. We thus deleted this sentence in the revision (in manuscript, Discussion: "Furthermore, the operation of relatively large capacity BIPP plants can potentially result in additional positive environmental benefit.").

Line 516/517: BECCS was said to be highly water intensive (due to amine capture technology?) but no information on the technology was provided to determine or compare the water use outlined here (cooling water usage was not given here either).

Response:

Thanks for the reviewer to point it out. We feel sorry that we did not make it clearly in the manuscript. Virtually, BECCS has higher water intensity compared to BIPP systems. Many researches indicated that the CCS component of BECCS was water intensive, and extra water was needed for the scrubbers that remove CO₂ from the air [1-3].

Smith et al. evaluated the water requirement for BECCS, the results show that for **additional water required for CCS would be 450 m³ t⁻¹ C_{eq} yr⁻¹** to the evaporative loss relative to bioenergy alone [1]. In this study, the main water use from BIPP system include cooling water usage and domestic water usage which can't be shown in Aspen simulation but have been considered in life cycle assessment. The cooling water usage in the studied BIPP plant is 4650 kg/h-t biomass feedstock under the assumed condition (i.e., 600°C with 7.5 t biomass feedstock input). The domestic water usage is assumed about 2460 kg/h as shown in feasibility report of BIPP plant. **We thus could calculate that there would be around 6.6 t water consumption**

for per t C negative emissions in one year ($6.6 \text{ m}^3 \text{ t}^{-1} \text{ C}_{\text{eq}} \text{ yr}^{-1}$), which is obviously lower than that of BECCS system.

Reference:

[1] Smith P, Davis S J, Creutzig F, et al. Biophysical and economic limits to negative CO₂ emissions [J]. Nature Climate Change, 2016, 6(1): 42-50.

[2] Heck V, Gerten D, Lucht W, et al. Biomass-based negative emissions difficult to reconcile with planetary boundaries [J]. Nature Climate Change, 2018, 8(2): 151-155.

[3] Rochelle G T. Amine scrubbing for CO₂ capture [J]. Science, 2009, 325(5948): 1652-1654.

Eqn (3): The authors should details the units for each of the variables used in equations. It is presumed CO₂ fixed is in kg while PBC is also kg, CBC and SBC are percentages and CF is MW of CO₂ divided by MW of C for a 1:1 molar conversion?

Response:

Thank the reviewer for pointing it out. We have revised the manuscript.

Corresponding change in text (in manuscript):

In Methods section:

Furthermore, the CO₂ fixation (CO₂ fixed with unit of t for one BIPP plant) in the biochar is calculated according to:

$$\text{CO}_2 \text{ fixed} = P_{BC} \times C_{BC} \times S_{BC} \times CF \quad (3)$$

where P_{BC} indicates the biochar yield from the pyrolysis system (unit: t/per plant), C_{BC} represents the fixed carbon content of biochar (unit: t fixed carbon for per t biochar), S_{BC} measures the stability rate (%) of carbon in soil, and CF defining the C-CO₂ conversion factor taken here as 3.67.

Line 658: It is unclear why 0%, 11% and 73% were chosen as scenario basis, and the ‘gradual’ rate of increase is also seemingly arbitrarily used? Justify further why these rates were used.

Line 683: Why 5%?

Response:

We appreciate the reviewer's comments. According to the current utilization of crop residues in China in 2014, biomass used as fuel accounts for about 11% of the total, as fertilizer about 35%, animal feed about 24%, industrial raw materials about 4% and base material about 4%. In addition to these five uses, around 22% of collected biomass is burned directly in the open air. Thus, in this study we assume that there is up to 11%, 33% (base case, the share currently used in energy systems and burned in fields), 73% (all except the biomass used in industry and for animal feed) and a maximum of 80% of available crop residues (all except crop residues used for animal feed assuming higher efficiency as developed countries have) that could be used for BIPP deployment with biochar sequestration. Constrained by the complex conditions (e.g., social, economic, environmental and political development) for BECCS deployment, the developing rate per year during 2030-2050 is really difficult to predict. Thus, in this study, we assume that the transition of BIPP plants to BECCS plants and deployment of BECCS in an annual average change rate: 11% crop residues used in BIPP plants in 2030 are transitioned to BECCS plants by 2050 with the annual average change rate of 0.74%; 33% crop residues are transitioned to BECCS plants with the rate of 2.7%; 73% crop residues are transitioned to BECCS plants with the rate of 13.0%.

Why 5%: It is calculated according to the assumption that 100% residues used in BIPP plants are transitioned to BECCS plants with an annual average change rate of 5% during 2030 to 2050 in scenario 6 and 7.

SUPPLEMENTARY DOCUMENT

Line 133: Fig S1 said to show the distribution of gaseous products of rapeseed straw. In fact Fig S1 is the ASPEN model and the gaseous product distribution cannot be found in figures presented, perhaps the authors mean Table S5?

Response:

Thank the reviewer for pointing out this error. Yes, it is Table S5. We have revised it.

Corresponding change in text (In Supplementary Information):

In **Model validation** section, page 4 line 130, “Fig. S1 shows the distribution of gaseous products of rapeseed residue at 650 °C” has been changed into “Table S5 shows the distribution of gaseous products of rapeseed residue at 650 °C”

Line 216: The value of bio-oils, heavy and light, is questionable from slow pyrolysis processes. The authors should justify their use of these values in more detail.

Response:

We appreciate the reviewer’s comments very much. We have justified the values of bio-oils (i.e. heavy and light oil) in Supplemental Information.

Corresponding change in text (in Supplementary Information):

2.1 Economic benefit from pyrolysis production

The prices for these products are collected from the Chinese trading market reported on the website of Alibaba, the feasibility report of the Ezhou BIPP plant¹¹ and a previous biomass feasibility report¹⁴; in the feasibility reports the prices are set by the relevant BIPP company according to purchase and sale agreements with other companies. The prices are listed below (Table S9).

Table S9. The price of every production in pyrolysis plant^{11,14}

Items	Price	Unit
Carbon price	4.54×10^2	USD/t
Biochar ^a	3.44×10^2	USD/t
Pyrolysis gas	$4.54 \times 10^{-2} / 6.81 \times 10^{-2}$	USD/Nm ³
Heavy oil	2.72×10^2	USD/t
Light oil	3.03×10^2	USD/t
Electricity	123×10^{-2}	USD/kWh

^a The price of biochar is used in sensitivity analysis.

Line 209-217: It is a strange assumption that the biochar has no value to the process. Considering the value of bio-oils is also variable and not fully established it surely will call into question why slow pyrolysis as considered at all. It is understood that the focus is mainly to

reduce greenhouse gas emissions but without a stronger economic driver for making biochar (compared to just doing fast pyrolysis for example and preferencing bio-oil formation) it seems unlikely that BIPP technology would be competitive. Surely the authors should also consider a case where biochar does have an economic value to demonstrate broader application and competitive uptake?

Response:

We would like to thank the reviewers for the helpful comments. The detailed response is as follows.

a. Firstly, we made an assumption that biochar has no value to the process for two reasons.

1) **Constrained by an undeveloped market for biochar and the lack of relevant policies for biochar production and sequestration in the soil:** At present, the research and development of biochar-based fertilizer has been supported by the National Key Research and Development Project in China, and has been included in the national rice, peanut, potato and other industrial technology systems. All of them could provide strong support for technological innovation of biochar utilization. However, many other problems that restrict the biochar industry development have not yet been resolved. For example, biochar-based fertilizers have not yet been included in the national new fertilizer category catalogue; carbon content in biochar-based fertilizers lacks cost-effective and standard testing methods; biochar-based fertilizers are relatively expensive.

2) **There is no profit from biochar from pyrolysis as assumed in previous research:**

Peters, J. F., Iribarren, D. & Dufour, J. Biomass pyrolysis for biochar or energy applications?

A life cycle assessment. *Environmental Science & Technology* **49**, 5195-5202 (2015).

However, we very much agree with the reviewer's valuable comments. Thus, in revision, we have considered the carbon price for BIPP system development, and discussed how it will change the BIPP system's economic performance in Sensitivity Analysis.

b. Secondly, we really agree with the reviewer's comment that the value of bio-oils is variable in the market. Firstly, it is believed that the value of bio-oils could be variable with the change of quality under different reaction condition. The light oil is mainly composed of

water, acids, esters, furans and phenols, the heavy oil is mainly composed of sugar-derived (levoglucan), free fatty acids and polycyclic aromatic hydrocarbons. However, due to the complexity and difficulty in bio-oils measurements, we assumed that the price of bio-oils (heavy and light oil) is the average price value in the BIPP company's *Purchase and Sale Agreements* made with other companies, which also lies in the actual price range in China trading market (Alibaba: <https://www.1688.com/>). In addition, it is believed that the value of bio-oils could be variable over time according to the fluctuation of crude oil price. **We also have considered the price value change of bio-oils in Sensitivity Analysis.**

Line 24 and 235: Discusses BIPP plants being built in locations where available biomass is abundant and easily supplied. This is an example of where a large assumption is applied in order to allow for a broad overarching model to be used. Perhaps this would be true in the case of low biomass utilisation scenarios, but would be highly unlikely in the case of the 80 or 100% usage case such as scenario 7?

Response:

We would like to thank the reviewer very much for this valuable comment. In “extreme” scenarios, changing the biomass transport distance can ensure the availability of feedstock for each BIPP plant. We thus have added a sensitive analysis about the impact of biomass transport distance on the net life-cycle assessment for one demonstration BIPP plant, and also on the cumulative GHG emission reduction during 2020-2050.

Corresponding change in text (in Supplementary Information):

Scenario F. Transport distance

For the base case scenario, BIPP plants are assumed to be built in locations where available biomass is abundant and easily supplied with an average transport distance of 40 km (based on the BIPP demonstration plant in case study ¹¹). However, in extreme situations assuming use of more than 80% of the China's biomass resources in BIPP systems, the transport distance would be expanded so as to satisfy the daily feedstock requirements from a certain BIPP plant. Thus, in sensitivity analysis, biomass transport distance changing from 20 km to 100 km is considered (shown in Table S27 and Table S30).

Table S30. Net life-cycle GHG emissions of different transport distance and models

Transportation		GHG emissions (t CO ₂ -eq)	Net life-cycle GHG emissions (g CO ₂ -eq/MJ)	Proportion in whole system (%)
20 km	Diesel	2.86×10 ²	-53.46	4.12
	Gasoline	4.06×10 ²	-52.59	5.74
	Hydrogen	2.12×10 ²	-54.00	3.08
	Electricity	8.51×10 ¹	-54.92	1.26
40 km	Diesel	5.73×10 ²	-51.38	7.92
	Gasoline	8.12×10 ²	-49.65	10.86
	Hydrogen	4.24×10 ²	-52.46	5.98
	Electricity	1.70×10 ²	-54.30	2.49
100 km	Diesel	1.43×10 ³	-45.15	17.69
	Gasoline	2.03×10 ³	-40.82	23.34
	Hydrogen	1.06×10 ³	-47.86	13.71
	Electricity	4.26×10 ²	-52.45	6.00

Line 335: Authors discuss the use of diesel vehicles. While this might be true now by 2050 surely this would be hydrogen or electric in nature? It would be interesting to have more detail in here including bringing in other predictions made around hydrogen usage in agriculture for example.

Response:

We would like to thank the reviewer very much for this valuable comment. With the carbon reduction for industries in the future, the hydrogen and electric vehicles would be excellent choices for transportation sector. In the revision, we have done a sensitivity analysis on transport model, considering hydrogen and electric vehicles.

For hydrogen usage in the future, it would be a very interesting discussion. There are many types of hydrogen technologies, including grey hydrogen, blue hydrogen and green hydrogen. However, considering the difficulty of collecting data for different hydrogen technologies, and complexity and uncertainty of prediction on future energy mix, we did not explore further here.

Corresponding change in text (in Supplementary Information):

6.2 Sensitivity analysis of the life-cycle GHG emissions for a demonstration BIPP system

Scenario G. Transport models

Based on the current status of vehicle utilization in China, the transport model of diesel vehicle is assumed in the base case life-cycle GHG emissions assessment. It is obvious that different transport models could lead to different GHG emissions. Hydrogen and electric vehicles, low-carbon transport options, may be feasible in the future, and thus have been considered in the sensitivity analysis. Table S29 shows the energy consumption of different transport models, and the resulting net life-cycle GHG emissions from different transport models are shown in Table S30.

6.3 Sensitivity analysis of modelling scenarios for China's bio-NETs deployment

Based on the results as mentioned above, for a single BIPP plant, the impact of N₂O reductions on the total life-cycle GHG emission is small (between -1.0% and 3.0%), while the variations in biochar stabilities and transport distance/models play an important role in the BIPP life cycle GHG emissions (with change rates of -19.7% to 34.6%, and -6.9% to 10.6%, respectively). Thus, the range of cumulative GHG reductions from a change in biochar stability and transport distance/models is studied further in several potential scenarios for future bio-NETs deployment in China.

The results in Fig. S11 show that the cumulative GHG reduction by 2050 could be affected to a certain extent (-3.9% to 5.9% relative to the base case) under the scenarios 1-4 and 6-7. Furthermore, it shows that under scenario 5 (assuming deployment of BIPP only), the cumulative GHG reduction by 2050 is more sensitive (-11.3% to 17.0% relative to base case), compared with other scenarios. In this sense, BIPP coordinated with BECCS deployment will result in a lower uncertainty in future cumulative GHG reduction by 2050, compared to deployment of BIPP alone (scenario 5).

For a mid-term perspective, the initial highest biomass input in BIPP coordinated with further BECCS deployment (scenario 4) can reach up to 8337-9127 Mt CO₂-eq reduction by the end of 2050. To put this in a worldwide context, the global emissions reduction target required to

meet the Paris Agreement goal of limiting the increase in global temperatures to no more than 1.5 °C by 2050 using BECCS technology was pegged at 28-65 Gt CO₂-eq⁵⁹. China by itself can achieve around 14-33% of the global removal goal by focusing strongly between now and 2050 on deployment of bio-NETs systems employing only agricultural, forest residues and energy plants (explained in SI 5.5). Furthermore, the scenario using 100% of agricultural residues, energy crops and forest residues (scenario 7) could provide cumulative reduction potentials of 21018 to 22674 Mt CO₂-eq by 2050. To sum up, these options under the “moderate” or “maximum” scenarios (aggressive deployment) could contribute to 175-3599 Mt CO₂-eq of GHG reductions by 2030. These value can be translated to a reduction on carbon emissions per unit of 2005 GDP by 2%-69%, which could approach or even achieve the goals alone (i.e., 60%-65% CO₂ reduction) included in China’s NDC for the Paris Agreement⁶³.

Line 355-357: Not enough information is provided about wastewater treatment. For example and amount of methane production annually is given but the amount of treated waste water generated by this is not supplied for reference.

Response:

We appreciate the reviewer’s comments very much. According to the reviewer’s comments, we have added the detailed information for the calculation wastewater treatment.

Corresponding change in text (in Supplementary Information):

3.2.1.4 Greenhouse gas emissions from the demonstration BIPP plant

In addition, the CH₄ emissions associated with the wastewater treatment are an important source of GHG emissions caused by the operation of a pyrolysis plant. In the wastewater treatment system, anaerobic processes lead to CH₄, N₂O and CO₂ emissions, although N₂O is too negligible to be considered and CO₂ is not considered because of its biogenic origin²⁵. According to the IPCC method²⁵, the equation for CH₄ emissions for wastewater treatment is as follows:

$$E_{CH_4} = \sum(TOW_i \cdot EF_i - R_i) \quad (8)$$

where E_{CH_4} represents the CH₄ emissions of the wastewater treatment; TOW_i represents the total organically degradable material in wastewater produced by industry i per year, expressed

in kg of chemical oxygen demand (COD)/yr; EF_i represents the emission factor for industry i , expressed in kg CH₄/kg COD; and R_i represents the amount of CH₄ recovered per year, expressed in kg CH₄/yr. At the time of the analysis year of 2018, China had not carried out large-scale recovery of CH₄, and as a result the default value of R_i is taken as zero. The CH₄ emission factor is therefore calculated as:

$$EF = B_0 \cdot MCF \quad (9)$$

where B_0 represents the maximum CH₄ producing capacity, expressed in kg CH₄/kg COD, with a default value of 0.25 kg CH₄/kg COD; and MCF stands for the methane correction factor, with a default value of 0.5, based on IPCC and domestic research^{25,26}.

In this study, the wastewater is mainly comprised of cooling water and domestic water. Cooling water is first used to decrease the temperature of the biochar, is then used in the second-stage separator (water cooling tower), and finally goes into the gas purification and absorption system. The second-stage separator is a shell-and-tube indirect heat exchanger. A condensate collection channel is arranged at the bottom of the heat exchanger, so light materials and part of the condensate water of pyrolysis gas can enter the light oil tank. In the gas absorption system, water is used to wash the pyrolysis gas so as to remove the contained trace tar. The final water output is wastewater and processed in a wastewater treatment system. Limited by the Aspen block and in order to simplify the system, this study has simulated the gas purification and absorption system (CYC-MIX, CYC-SPI, CYC-PUM and SPR-SEP) without washing water flow and the wastewater treatment system. Domestic water is daily water consumption by workers, which is ignored in the system diagram.

Nevertheless, the volume of the treated wastewater (cooling water and domestic water) is calculated based on the feasibility report. Based on available data and equations, the CH₄ emissions associated with wastewater treatment are 37.0 t/yr (924 t CO₂-eq/yr).

Table S14. The CH₄ emissions associated with wastewater treatment for a BIPP system with 7.5 t biomass feedstock input per hour

Wastewater items	Quantity (kg/h)	COD (kg/L) ^a	CH ₄ emissions (t/yr)
Cooling water	4650	0.0125	36.5
Domestic water	2460 ^a	0.0003	0.5
Total	7110	-	37.0

^b From feasibility report¹¹

Line 360: 'in used' should be deleted.

Response:

Thank the reviewer for pointing it out. We have revised the sentence.

Line 477-478: Should justify the assumption that pyrolysis technology (BIPP specifically) will allow removal of particulates mercury etc. When combusting biomass for heat inputs particulates will certainly be generated and must be treated. How can this be controlled better than existing plants also using combustion?

Response:

We appreciate the reviewer's comments very much.

Short answer:

In this study, we did not consider using extra method to control emissions of particulates mercury in the BIPP demonstration plant, as the mercury emission intensity of BIPP system (including combusting biomass for heat inputs) is significantly lower than China's emission standards. Nevertheless, we agree with the reviewer that particulates will certainly be generated and must be treated. In our recent experimental study, we found that biochar could be used to reduce particulates. We will consider it in our future work.

In-detail explanation is as follows:

According to the *Emission Standard of Air Pollutants for Thermal Power Plants* (GB13223-2011, implementation in 2015 in China) and *Boiler Air Pollutant Emission Standard* (DB37/2374-2018, in China), the emission concentration limit of mercury and its compounds is **0.03 mg/m³ and 0.05 mg/m³**, respectively. The average emission factors (EFs) of total mercury from eight kinds of crop residues combustion is 6.54 ± 3.47 ng/g; the average mercury EFs from 17 kinds of firewood combustion is 9.27 ± 8.41 ng/g [1]. Based on the Aspen simulation, there would be 0.29 kmol air needed (0.18 kmol stoichiometric air for mixed crop residues shown in Table S2, 60% excess air) and 8.34 m³ emitted fuel gas (output temperature is about 43 °C) for 1 kg biomass fuel combustion in BIPP system. Based on the

above mentioned data, it can be estimated that the possible mercury emission for crop residues combustion in this BIPP system is $0.00078 \pm 0.00043 \text{ mg/m}^3$, and for firewood combustion is $0.0011 \pm 0.0010 \text{ mg/m}^3$, which **are significant lower than China's emission standards**. In addition, the Ezhou demonstration BIPP plant does not equip with the mercury removal equipment, thus, the Aspen simulation in this system also did not consider this kind of equipment such as popular wet flue gas desulfurization (WFGD) in China's thermal power plant.

However, we totally agree with the reviewer's comment that there would be particulates removal equipment in future, if the BIPP system would be deployed widely in China. At present, the main technologies used to remove mercury in China's fossil-fuel power plant include selective catalyst reduction (SCR) and flue gas desulfurization (FGD) [2], which target on air pollution control (i.e. denitrification and desulfurization, respectively). The SCR catalysts could promote the oxidation of some of the Hg^0 to Hg^{2+} and FGD system could remove some of Hg^{2+} . The coal-fired power plants in China are generally equipped with SCR+ESP (electrostatic precipitators, Hg^{P} absorbed in fine particulate matter capture) +WFGD or ESP+SCR+WFGD. Another relatively mature technology for mercury removal is activated carbon adsorption, which would be expensive in industrial applications. And all these technologies mentioned above is not cost-effective when applied in BIPP systems, which have a relatively low capacity and low mercury emission intensity compared with coal-fired power plants. In recent researches, using biochar from biomass pyrolysis as mercury adsorption (especially Hg^{2+}) has attracted large attention [3-5]. Thus, with the development of BIPP in the future, biochar (after upgrading) may be used in particulate removal for biomass combustion. In our recent experimental study, we found that biochar could be used to reduce particulates [6-7]. We think it would be an interesting topic, and will discuss these possible scenarios in future research.

Reference:

- [1] Wei W. Emission of mercury from biomass fuels burning in rural China [D]. Peking University, 2012.
- [2] Liu K, Wang S, Wu Q, et al. A highly resolved mercury emission inventory of Chinese coal-fired power plants [J]. Environmental Science & Technology, 2018, 52(4): 2400-2408.

- [3] Zhang H, Wang T, Sui Z, et al. Plasma induced addition of active functional groups to biochar for elemental mercury removal[J]. *Plasma Chemistry and Plasma Processing*, 2019, 39(6): 1449-1468.
- [4] Liu D, Li C, Wu J & Liu Y. Novel carbon-based sorbents for elemental mercury removal from gas streams: A review [J]. *Chemical Engineering Journal*, 2019, 123514.
- [5] Xu W, Adewuyi Y G, Liu Y & Wang Y. Removal of elemental mercury from flue gas using CuO_x and CeO₂ modified rice straw chars enhanced by ultrasound. *Fuel Processing Technology*, 2018, 170, 21-31.
- [6] Zhang S, Zhang H, Cai J, et al. Evaluation and prediction of cadmium removal from aqueous solution by phosphate-modified activated bamboo biochar[J]. *Energy & fuels*, 2017, 32(4): 4469-4477.
- [7] Liao X, Shao J, Zhang S, et al. Effects of CO₂ and CO on the reduction of NO over calcined limestone or char in oxy-fuel fluidised bed combustion[J]. *IET Renewable Power Generation*, 2019, 13(10): 1633-1640.

Table S5: Would be useful to have a standard deviation or idea of error in experimental results as this is known to have some variance and could then confirm the simulation results are within the range of experimental error.

Response:

Thank the reviewer to point it out. We have added the errors of experimental results in Table S5, and error bars in Fig. S2.

Corresponding change in text (in Supplementary Information):

Table S6. The gas component distribution of rapeseed pyrolysis at 650 °C

Component	Experiment (kmol/t biomass)	Errors (kmol/t biomass)	Simulation (kmol/t biomass)
CO	3.5	0.29	3.2
CO ₂	2.1	0.23	2.6
H ₂	2.7	0.29	2.1
C ₂ H ₄	0.4	0.34	0.7
CH ₄	2.2	0.26	1.8

Fig. S2. The comparison of Aspen Plus simulation results and experimental data as a function of temperature